# Functional and structural analyses reveal that a dual domain sialidase protects bacteria from complement killing through desialylation of complement factors

**Nicholas D. Clark**[1�], **Christopher Pham**[2�], **Kurni Kurniyati**[2], **Ching Wooen Sze**[2], **Laurynn Coleman**[2], **Qin Fu**[3], **Sheng Zhang**[3], **Michael G. Malkowski**[1]*, **Chunhao Li**[2]*

1 Department of Structural Biology, Jacobs School of Medicine and Biomedical Sciences, University of Buffalo, the State University of New York, Buffalo, New York, United States of America, 2 Philips Institute for Oral Health Research, Virginia Commonwealth University, Richmond, Virginia, United States of America, 3 Proteomics Facility, Institute of Biotechnology, Cornell University, Ithaca, New York, United States of America

l These authors contributed equally to this work.

\* mgm22@buffalo.edu (MGM); cli5@vcu.edu (CL)

**Data Availability Statement:** The authors confirm that all data underlying the findings are fully available without restriction. All relevant data are

## Abstract

The complement system is the first line of innate immune defense against microbial infections. To survive in humans and cause infections, bacterial pathogens have developed sophisticated mechanisms to subvert the complement-mediated bactericidal activity. There are reports that sialidases, also known as neuraminidases, are implicated in bacterial complement resistance; however, its underlying molecular mechanism remains elusive. Several complement proteins (e.g., C1q, C4, and C5) and regulators (e.g., factor H and C4bp) are modified by various sialoglycans (glycans with terminal sialic acids), which are essential for their functions. This report provides both functional and structural evidence that bacterial sialidases can disarm the complement system via desialylating key complement proteins and regulators. The oral bacterium *Porphyromonas gingivalis*, a "keystone" pathogen of periodontitis, produces a dual domain sialidase (PG0352). Biochemical analyses reveal that PG0352 can desialylate human serum and complement factors and thus protect bacteria from serum killing. Structural analyses show that PG0352 contains a N-terminal carbohydrate-binding module (CBM) and a C-terminal sialidase domain that exhibits a canonical six-bladed β-propeller sialidase fold with each blade composed of 3–4 antiparallel β-strands. Follow-up functional studies show that PG0352 forms monomers and is active in a broad range of pH. While PG0352 can remove both N-acetylneuraminic acid (Neu5Ac) and N-glycolyl-neuraminic acid (Neu5Gc), it has a higher affinity to Neu5Ac, the most abundant sialic acid in humans. Structural and functional analyses further demonstrate that the CBM binds to carbohydrates and serum glycoproteins. The results shown in this report provide new insights into understanding the role of sialidases in bacterial virulence and open a new avenue to investigate the molecular mechanisms of bacterial complement resistance.

within the paper and its Supporting Information files.

**Funding:** This project is supported by F31 fellowship award (DE029999) to C.P.; USPHS award GM095459 to N.D.C.; DE030667 to C.L., and DE023080 to C.L. and M.G. M. The funders had no role in study design, data collection and analysis, decision to publish, or preparation of the manuscript.

**Competing interests:** The authors have declared that no competing interests exist.

## Author summary

The human complement pathway is a highly efficient recognition and effector system dedicated to destroy infectious microbes and damaged host materials and thus it has been considered as the first line of innate immune defense. Bacterial pathogens have evolved various tactics to evade the complement system and cause infections. Complement resistance mechanisms can serve as novel therapeutic targets for defending against highly antibiotic-resistant pathogenic bacterial infections, including periodontitis, a chronic inflammatory illness that is triggered by polymicrobial infection. Therefore, understanding the interplay between the complement system and bacterial pathogens is critical for developing new therapeutics against bacterial infections including periodontitis. Bacterial sialidases have been reported to be implicated in complement resistance, and yet its underlying mechanism remains largely unknown. This report reveals that sialidases confer bacteria complement resistance via desialylating key complement proteins and regulators, which can potentially serve as a new therapeutic target for treatment of infectious diseases.

## Introduction

The human complement system is one of the critical defense mechanisms of innate immunity [1,2]. It is adept in the recognition and clearance of foreign pathogens and prevents microbial infections. The complement system can be activated by antigen-antibody complexes (classical pathway, CP), certain carbohydrates (lectin pathway, LP), or by a variety of surfaces that are not protected with natural inhibitors (alternative pathway, AP) [1,3]. CP is initiated when the C1q recognition protein of the C1 complex binds to an antigen-antibody complex [4]. AP is activated by continuous "turnover" of C3 into C3b which can bind to foreign pathogens in a process called opsonization. LP is triggered by binding of mannose-binding lectin (MBL) to carbohydrate residues on pathogen surfaces. All three pathways converge in formation of a C3 convertase which triggers an activation cascade culminating in formation of the membrane attack complex (MAC) onto bacterial cell surfaces. The MAC forms cytotoxic pores that disrupt bacterial cell membranes and result in cell lysis and death. In addition to the innate immunity, the complement system is also interwoven with the adaptive immunity, e.g., opsonization by the complement system can interact with various receptors on B cells (i.e., Complement receptor 1 and 2) and modulate their responses [1,5]. Likewise, the complement system can also interface with T cells, i.e., complement deficiency reduces priming of CD4 and CD8 T cells [6].

The complement system is tightly regulated to prevent undesired destruction of healthy host cells. There are several fluid-phase regulators for the complement system, such as factor H (FH), factor I (FI), C4-binding protein (C4bp), and properdin [3,7]. FH is a soluble mediator that negatively regulates the complement system. It recognizes and binds to structures found on host cells to prevent collateral damages [8]. FH also functions as a cofactor for FI, a serine protease that cleaves C3b to inactivate the complement system [9]. C4bp inhibits complement activation by binding C4 within the C3 convertase to promote its dissociation, thereby inhibiting the activation cascade [10]. C4bp can also function as a cofactor for FI in a similar manner to FH. Properdin functions to upregulate complement activation. It binds and stabilizes the C3 convertase while protecting the convertase against FH and FI mediated cleavage [11].

To establish infection and persist in human hosts, bacterial pathogens must successfully evade the complement system as it is the first line of defense of the immune system [12]. Pathogenic bacteria have evolved sophisticated mechanisms to subvert complement-mediated killing (also referred to as serum killing), including disrupting complement factors with proteases, hijacking host complement regulators such as FH to prevent complement activation, 'hiding' within host cells, either in a vacuole or in the cytoplasm, or employing a physical barrier such as polysaccharide capsules to prevent the complement activation and deposition [for review, see [12–14]]. For example, *Neisseria gonorrhoeae* captures circulating serum FH to inhibit complement activation [15,16]. *Borrelia burgdorferi*, the causative agent of Lyme disease, confers its complement resistance using complement regulator-acquiring surface proteins (CRASPs) which can recruit host FH and FH-like protein to downregulate the complement system [17]. Its surface adhesin molecule BBK32 inhibits the CP activation by blocking activation of the C1 complex [18,19] and outer surface protein C (OspC) blocks the CP and LP activation through competition with C2 for C4b binding [20]. *Streptococcus pyogenes* produces exotoxin B, a cysteine protease, that degrades C3 thereby inhibiting complement activation [21]. Similarly, a cysteine proteinase from *Prevotella intermedia* inhibits complement by degrading C3 [22]. *S. pneumoniae* capsule confers its complement resistance through blocking conversion of C3b to iC3b and its subsequent opsonization on bacterial cell surfaces [23]. There are also reports that bacterial sialidases (also known as neuraminidases) are implicated in complement resistance. For example, the NanA sialidase of *S. pneumoniae* confers its serum resistance by blocking activation of the AP [24]. Our recent studies also revealed that sialidase-deficient mutants of *Treponema denticola* and *Porphyromonas gingivalis*, two "keystone" pathogens of human periodontitis [25], became susceptible to complement killing [26,27]. However, the molecular mechanism by which sialidases protect bacteria from complement killing remains largely unknown.

The complement system consists of over 30 proteins, including plasma proteins, regulatory factors, receptors, and ligands, and most of them are modified by various O-linked and N-linked glycans, which are critical for their functions, stability, protein-protein interactions, and self-recognition [7]. Interestingly, the majority of those glycans contain terminal N-acetylneuraminic acids (Neu5Ac), the most common form of sialic acids in humans. For example, C1q, C4, C5, FH, and FI are all modified by various sialoglycans at multiple sites [7,28]. Sialylation is believed to minimize non-specific interactions of the C1 complex by stabilizing C1q [7]. Removing sialic acid (desialylation) from FH can alter its function and lead to pathogenic effects [29]. In this report, using functional and structural approaches, we demonstrate that PG0352, a sialidase that was previously identified in *P. gingivalis* W83 strain [27], can remove Neu5Ac from human serum and several key complement factors such as C1q, C4, C5, FH, and FI, and thus surrender serum bactericidal activity. In addition, we present the crystal structures of apo PG0352 and ligands- and inhibitor-bound PG0352. The results shown in this report provide new insights into understanding the role of bacterial sialidases and open a new avenue to investigate the molecular mechanisms of bacterial complement resistance.

## Results

### PG0352 removes human serum sialic acids

Human serum contains a high concentration of sialic acids that are conjugated to various glycoproteins, some of which contribute to complement activation and its bactericidal activity [30,31]. *P. gingivalis* (*Pg*) is resistant to serum killing; however, this resistance is abolished when the gene encoding PG0352 was deleted in the W83 strain [27]. We reasoned that PG0352 may protect *Pg* from serum killing via removing sialic acids from serum glycoproteins.

To test this hypothesis, we treated human serum with recombinant PG0352 followed by detection of sialic acid using *Sambucus nigra* (SNA) lectin, which specifically recognizes α-2,3- and α-2,6-linked sialic acids. For this experiment, human serum was incubated with varying amounts of wild-type PG0352 as labeled in Fig 1. In addition, a mutant PG0352 construct in which residues Tyr-193, Arg-194, Ile-195 and Pro-196, four residues that constitute a highly conserved 'F/YRIP" motif that is essential for bacterial sialidases [32], were collectively mutated to alanine generating an inactive enzyme, PG0352^AAAA, which was included as a negative control. After the incubation, samples were collected at 10, 30, and 60 minutes and subjected to lectin blot analysis. SNA detected several protein bands in human serum which were gradually abolished by the treatment with wild-type PG0352, but not PG0352^AAAA, in a time- and dose-dependent manner (**Fig 1**). By 60 minutes, all the signals detected by SNA disappeared. In contrast, the samples treated with PG0352^AAAA remained unchanged. To rule out the possibility that the treatment might lead to protein degradation, the samples treated for 60 minutes were subjected to lectin blots utilizing Concanavalin A (ConA) lectin, which detects internal core mannose residues. Treatment with wild-type PG0352 had no impact on ConA signals across all the samples, indicating that the protein does not affect the stability of serum proteins (**Fig 1**). Taken together, we conclude that PG0352, as a sialidase, can desialylate human serum glycoproteins.

## PG0352 desialylates complement proteins

Several complement factors are glycosylated, some of which are sialylated, e.g., C1q is modified by six-N-linked glycans, five of which are sialoglycans [7] and FH is modified by N-linked glycans at 7 sites and at least three of these sites are modified by sialoglycans [7,33]. Based on the above result, we reasoned that PG0352 could desialylate complement proteins which constitute a major part of serum. To test this hypothesis, we examined a panel of glycosylated complement proteins, including C1q, C3, C4, C5, IgG, FH, FI, and C4bp (**Fig 2**). Notably, C3 is glycosylated but not sialylated [7] which was included as a control. These proteins were incubated with wild-type PG0352, PG0352^AAAA or *C. perfringens* sialidase (NanI) at 37˚C for 30 minutes, except for IgG where treatment was extended to overnight (**Fig 2D**). Treated complement proteins were subjected to SDS-PAGE and SNA lectin blots (**Fig 2**). As expected, there was no sialic acid signal detected in C3 because it is glycosylated but not sialylated, highlighting the substrate specificity of SNA (**Fig 2A**). There were sialic acid signals detected in C1q, C4, C5, FH, FI, and C4bp, which were either completely abolished or substantially decreased by wild-type PG0352, but not PG0352^AAAA (**Fig 2**). IgG was more resilient to the treatment and took overnight incubation to completely remove sialic acid signals (**Fig 2D**). Interestingly, we found that PG0352 was much more active than the NanI sialidase of *C. perfringens* in terms of releasing sialic acid from these complement proteins (**Fig 2B–2H**). This difference could be due to their structure differences or other unknown factors. *C. perfringens* produces three different sialidases (i.e., NanH, NanI, and NanJ) which have different activity, substrate specificity, and functions [34,35]. Herein, we only tested NanI [36]. We are in the process of testing the other two sialidases against human serum and complement factors. Notably, SDS-PAGE analysis showed that the treatment with PG0352 has no impact on the stability of these complement proteins (**S1 Fig**). Collectively, these results indicate that PG0352 specifically desialylates these tested complement factors without impacting their stability.

## Mapping the glycosylation sites and glycans of human FH

The above biochemical analyses showed that PG0352 could remove sialic acids from serum proteins and several complement factors. To substantiate this study, we used FH as a surrogate

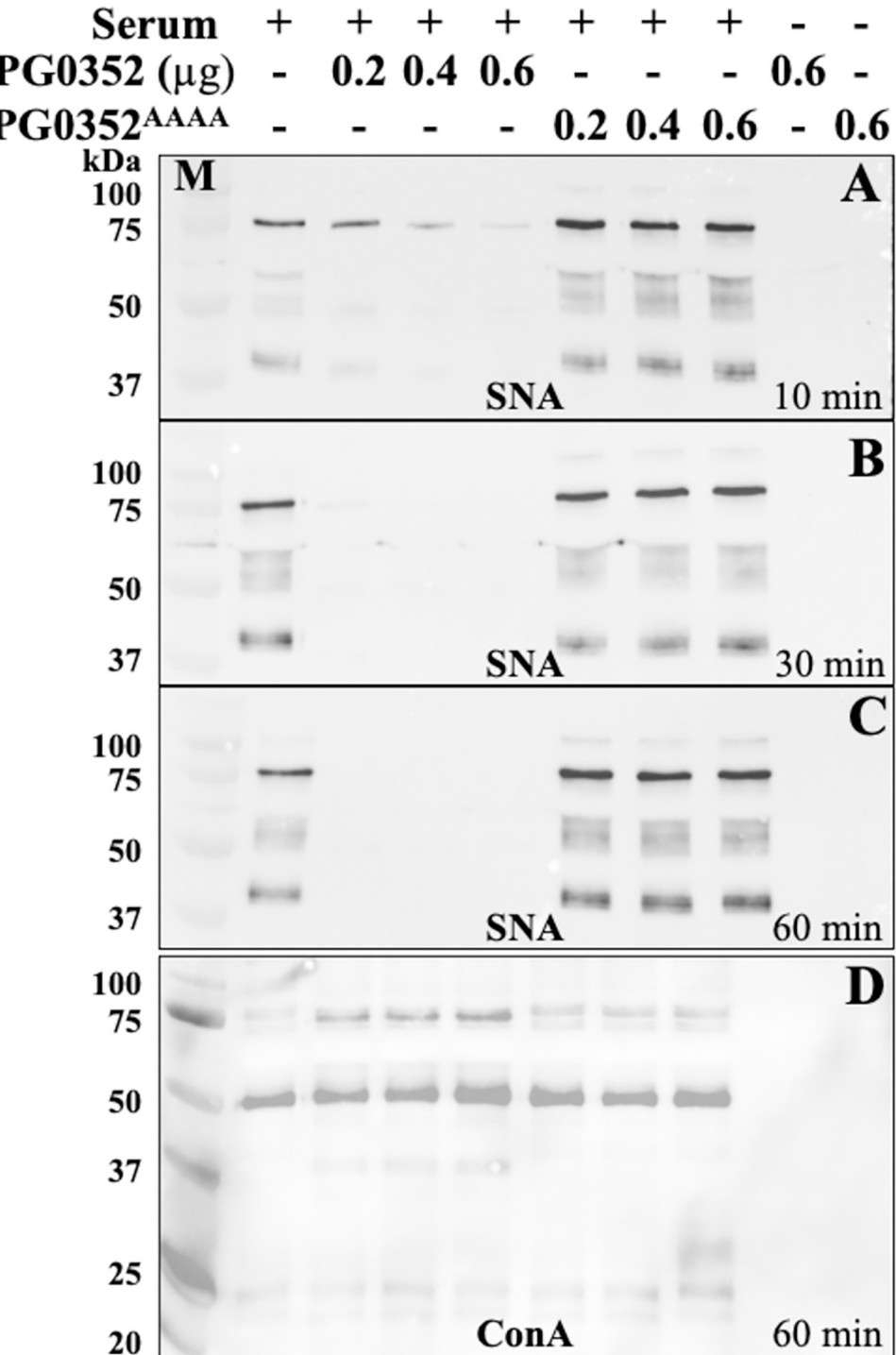

**Fig 1. PG0352 Desialylates Normal Human Serum.** Normal human serum was treated with 0.2 ~0.6 µg of wild-type PG0352 or inactive PG0352 (PG0352[AAAA]) as described in the Methods section. Samples were collected and subjected to SDS-PAGE and lectin blot analysis utilizing *Sambucus nigra* (SNA) and Concanavalin A (ConA) lectins **(A)** 10 minutes post-treatment with SNA; **(B)** 30 minutes post-treatment with SNA; **(C)** 60-minutes post-treatment with SNA; **(D)** 60-minutes post-treatment with ConA.

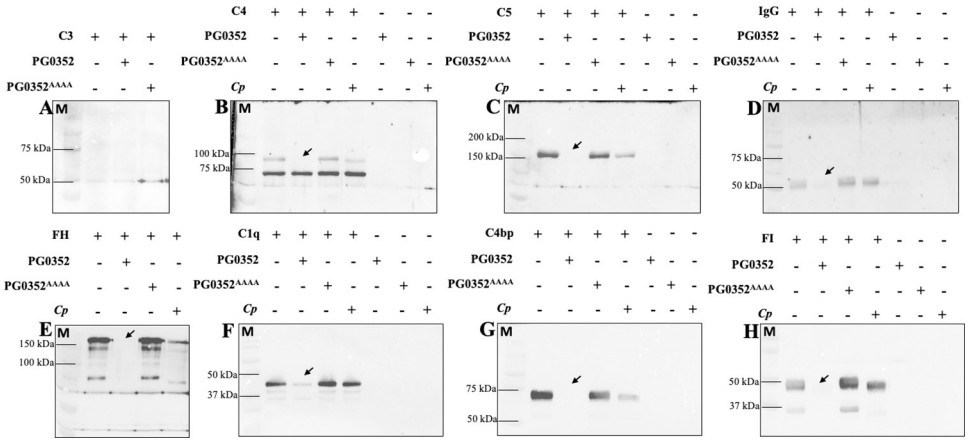

**Fig 2. PG0352 Desialylates Different Complement Factors.** Individual complement factors were treated with wild-type PG0352, inactive PG0352 (PG0352^AAAA) or sialidase from *C. perfringens* (*Cp*) as described in the Methods section. Samples were collected and subjected to SDS-PAGE and lectin blot analysis utilizing SNA lectin. **(A)** C3, **(B)** C4, **(C)** C5, **(D)** IgG, **(E)** FH (factor H), **(F)** C1q, **(G)** C4bp (C4 binding protein), and **(H)** FI (factor I). Arrows pointed to the bands are desialylated and could not be detected by SNA after incubation with recombinant PG0352.

to further assess the desialylation activity of PG0352 because this protein is heavily glycosylated and sialylated [33]. For this experiment, human FH protein was incubated with or without PG0352 for 3 hours at 37°C. The resulting samples were subjected to SDS-PAGE and then stained with Coomassie blue. As shown in Fig 3A, after the treatment with PG0352, the size of FH became smaller compared to untreated FH, suggesting that desialylation occurred. To further confirm this result, the protein bands pertaining to untreated (band A) and treated (band B) FH were excised (**Fig 3A**) and subjected to nano-LC-ESI/MS/MS for mapping its glycosylation sites and glycans. In the untreated FH sample, 4 glycosylation sites (N217, N882, N911, and N1029) and 13 sialoglycans were confidently detected in FH (**Table 1**). The same glycosylation sites were detected in the FH treated with PG0352, but none of those pertaining glycans contain terminal Neu5Ac (**Table 1**). A representative LC-ESI-MS/MS spectrum of FH glycopeptide with or without treatment of PG0352 was shown in Fig 3. We repeated this experiment using C1q and a similar pattern was observed (**S2 and S3 Figs, and S2 and S3 Tables**). Collectively, these results indicate that PG0352, as a sialidase, can remove terminal sialic acid from sialylated complement proteins.

## PG0352 disarms serum killing against susceptible bacteria

Our previous study shows that *ΔPG0352*, a deletion mutant of *PG0352*, is susceptible to complement killing while its parental wild type strain is resistant [27]. Since PG0352 can desialylate key complement factors such as C1q, C4, and C5, we speculated that this activity may directly protect *Pg* from the complement killing. To test this speculation, we first conducted 'extracellular' complementation assays by adding recombinant PG0352 to *ΔPG0352* prior to challenging with serum. However, we found that the recombinant protein was rapidly degraded most likely by gingipains, a group of proteases produced by *Pg* [37–39]. We also tried to complement the mutant using the inactive form of PG0352 constructed above but without success. To overcome these technique issues, we performed protection assays using *E. coli* NEB5α, a non-pathogenic strain that is susceptible to serum killing [40], as a surrogate. For this experiment, human serum samples, diluted to 2% and 5%, were pre-treated with either wild-type PG0352 or PG0352^AAAA and the resulting samples were used for serum killing assays. After a

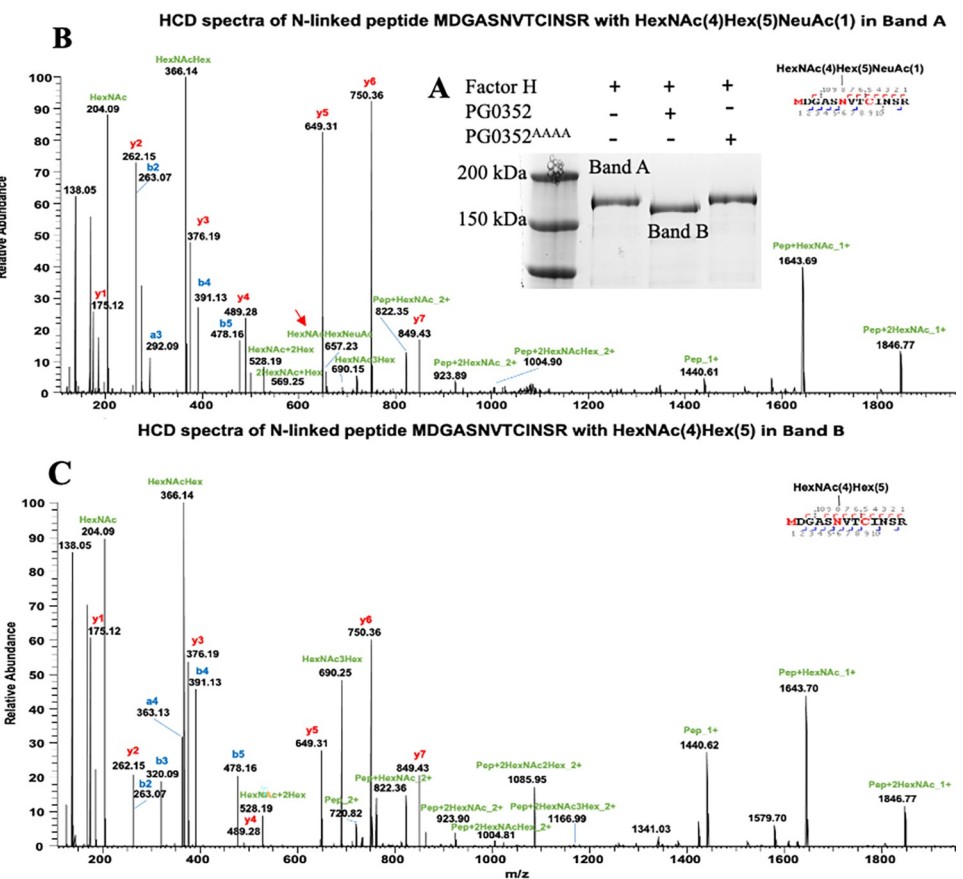

**Fig 3. PG0352 desialylates human Factor H (FH). (A)** SDS-PAGE of FH untreated (band A) and treated (band B) with recombinant PG0352 or PG0352[AAAA] for 3 hours at 37˚C. Bands A and B were excised for nano-LC-ESI/MS/MS to examine N-linked glycans. **(B)** & **(C)** A representative LC-ESI-MS/MS spectrum of FH glycopeptide untreated (B) and treated (C) with PG0352.

60-minute incubation, the average survival rate of *E. coli* in 2% human serum treated with PG0352 was ~93%, which dropped to 47% in the sample treated with PG0352[AAAA] (**Fig 4A**). A similar pattern was observed in the samples treated with 5% human serum, where the survival rate of *E. coli* dropped from 78.67% to 42% (**Fig 4B**). To substantiate these findings, we assessed complement deposition on *E. coli* cells to determine if PG0352 confers the protection through inhibition of complement activation and subsequent formation of MAC (C5b-9) on bacterial cell surfaces. As expected, complement factors, such as C3, C5-9, and C9, were detected in the *E. coli* samples treated with 5% normal human serum. By contrast, only trace amounts of C3, C5-9 and C9 signals were detected in those samples treated with PG0352 (**Fig 4C**). These results demonstrate that the pretreatment of human serum with PG0352, but not its inactive form, can protect bacteria such as *E. coli* from serum killing, most likely through desialylation of key complement factors such as C1q, C4, and C5.

## Crystal structure of unliganded PG0352

To elucidate the catalytic mechanism of PG0352, we first determined its unliganded crystal structure using synchrotron radiation to 1.84Å in a monoclinic space group. Initial phases were obtained utilizing molecular replacement in conjunction with a model of the protein generated using AlphaFold2. The overall structure of PG0352 consists of two major domains

**Table 1. MS/MS spectra for nano-LC/MS/MS for human Factor H (FH) before and after the treatment of PG0352.**

| Untreated Factor H Band | | PG0352 Treated Factor H Band | |
|---|---|---|---|
| Peptide Sequences | Identified Glycans | Peptide Sequences | Identified Glycans |
| K.SPDVIN[+2204.77244]GSPISQK.I | HexNAc(4)Hex(5)NeuAc(2) | K.SPDVIN[+1622.58161]GSPISQK.I | HexNAc(4)Hex(5) |
| K.IPC[+57.02146]SQPPQIEHGTIN[+2861.00005]SSR.S | HexNAc(5)Hex(6)NeuAc(3) | K.IPC[+57.02146]SQPPQIEHGTIN[+2133.77171]SSR.S | HexNAc(5)Hex(6)Fuc(1) |
| K.IPC[+57.02146]SQPPQIEHGTIN[+2350.83035]SSR.S | HexNAc(4)Hex(5)Fuc(1)NeuAc(2) | K.IPC[+57.02146]SQPPQIEHGTIN[+1987.71380]SSR.S | HexNAc(5)Hex(6) |
| K.IPC[+57.02146]SQPPQIEHGTIN[+2204.77244]SSR.S | HexNAc(4)Hex(5)NeuAc(2) | K.IPC[+57.02146]SQPPQIEHGTIN[+1768.63952]SSR.S | HexNAc(4)Hex(5)Fuc(1) |
| K.IPC[+57.02146]SQPPQIEHGTIN[+2059.73493]SSR.S | HexNAc(4)Hex(5)Fuc(1)NeuAc(1) | K.IPC[+57.02146]SQPPQIEHGTIN[+1622.58161]SSR.S | HexNAc(4)Hex(5) |
| K.IPC[+57.02146]SQPPQIEHGTIN[+1913.67702]SSR.S | HexNAc(4)Hex(5)NeuAc(1) | | |
| R.ISEEN[+2861.00005]ETTC[+57.02146]YM[+15.99492]GK.W | HexNAc(5)Hex(6)NeuAc(3) | R.ISEEN[+2133.77171]ETTC[+57.02146]YM[+15.99492]GK.W | HexNAc(5)Hex(6)Fuc(1) |
| R.ISEEN[+2221.78776]ETTC[+57.02146]YM[+15.99492]GK.W | HexNAc(4)Hex(6)Fuc(1)NeuAc(1) | R.ISEEN[+1987.71380]ETTC[+57.02146]YM[+15.99492]GK.W | HexNAc(5)Hex(6) |
| R.ISEEN[+2204.77244]ETTC[+57.02146]YM[+15.99492]GK.W | HexNAc(4)Hex(5)NeuAc(2) | R.ISEEN[+1622.58161]ETTC[+57.02146]YM[+15.99492]GK.W | HexNAc(4)Hex(5) |
| R.ISEEN[+1913.67702]ETTC[+57.02146]YM[+15.99492]GK.W | HexNAc(4)Hex(5)NeuAc(1) | | |
| K.M[+15.99492]DGASN[+2350.83035]VTC[+57.02146]INSR.W | HexNAc(4)Hex(5)Fuc(1)NeuAc(2) | K.M[+15.99492]DGASN[+1622.58161]VTC[+57.02146]INSR.W | HexNAc(4)Hex(5) |
| K.M[+15.99492]DGASN[+2204.77244]VTC[+57.02146]INSR.W | HexNAc(4)Hex(5)NeuAc(2) | | |
| K.M[+15.99492]DGASN[+2059.73493]VTC[+57.02146]INSR.W | HexNAc(4)Hex(5)Fuc(1)NeuAc(1) | | |
| K.M[+15.99492]DGASN[+2059.73493]VTC[+57.02146]INSR.W | HexNAc(4)Hex(5)Fuc(1)NeuAc(1) | | |
| K.M[+15.99492]DGASN[+1913.67702]VTC[+57.02146]INSR.W | HexNAc(4)Hex(5)NeuAc(1) | | |
| K.M[+15.99492]DGASN[+1913.67702]VTC[+57.02146]INSR.W | HexNAc(4)Hex(5)NeuAc(1) | | |
| R.N[+3663.28558]YSYEK.L | HexNAc(6)Hex(7)Fuc(1)NeuAc(4) | R.N[+2425.88753]YSYEK.L | HexNAc(5)Hex(6)Fuc(3) |
| R.N[+3155.13152]YSYEK.L | HexNAc(7)Hex(8)Fuc(1)NeuAc(1) | R.N[+2311.81945]YSYEK.L | HexNAc(5)Hex(8) |
| R.N[+3083.11039]YSYEK.L | HexNAc(8)Hex(9) | | |
| R.N[+3009.07361]YSYEK.L | HexNAc(7)Hex(8)NeuAc(1) | | |

encompassing 493 residues (31–523) (**Fig 5** and **Table 2**). The small N-terminal domain (residues 31 through 176) is a putative carbohydrate binding module (CBM), while the C-terminal domain (residues 177–523) exhibits a canonical six-bladed β-propeller sialidase fold, with each blade composed of 3–4 antiparallel β-strands. The overall structure of unliganded PG0352 is very similar to a recent report (41).

The CBM consists of two parallel β-sheets comprised of three β-strands on one face and five β-strands on the opposite face (**Fig 5**). The β-sheets form a concave surface which form an open cleft that is oriented towards the active site of the sialidase domain. A DALI search [42] utilizing the structure of the CBM identified Sialidase26 found in gut bacteria (PDB entry 6MRV) [43] as the closest structural homolog, with a sequence identity of 16.9%, Z-score of 12.2 and a root mean square deviation (RMSD) of 2.18Å for 127 equivalent Cα atoms. The recently characterized NanH sialidase from *Tannerella forsythia* (*Tf*NanH; PDB entry 7QYJ)

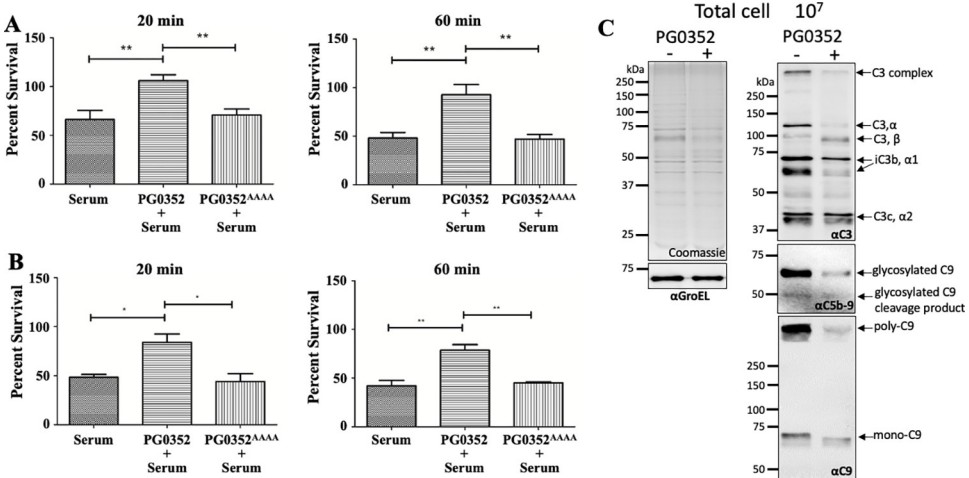

**Fig 4. PG0352 Inhibits the Bactericidal Activity of Serum.** Serum killing assays were conducted on *E. coli* NEB5α using normal human serum pre-treated with wild-type PG0352 or its inactive form PG0352[AAAA], as described in the Materials and Methods section. Heat-inactivated serum was used as a control. **(A)** 2% serum; **(B)** 5% serum. Statistical analysis was determined using one-way ANOVA followed by Tukey's multiple comparison at $P < 0.05$. **(C)** Complement deposition assays. A total of $10^7$ cells of *E. coli* cells were co-incubated with 5% normal human serum (-) or PG0352 treated (+) serum for 20 minutes at 37°C. The resultant serum-treated cells were harvested, washed, and resuspended in 50 μl PBS. Approximately 10 μl samples were subjected to SDS-PAGE, followed by immunoblotting analysis with four different antibodies as labelled: αC3 (a polyclonal antibody against C3), αC9 (a monoclonal antibody against C9), αC5-9 (a monoclonal antibody against C5-9 complex), and αGroEL (a polyclonal antibody against *E. coli* GroEL). These antibodies were purchased from Abcam.

[44] also contains an N-terminal domain that displays strong structural homology to the CBM, despite exhibiting a sequence identity of 16.6%. As the CBM of both Sialidase26 and *Tf*NanH belong to the CBM93 family, as defined in the Carbohydrate Active Enzymes (CAZy) database [45], it is likely that the CBM of PG0352 is a member as well. It is postulated that the

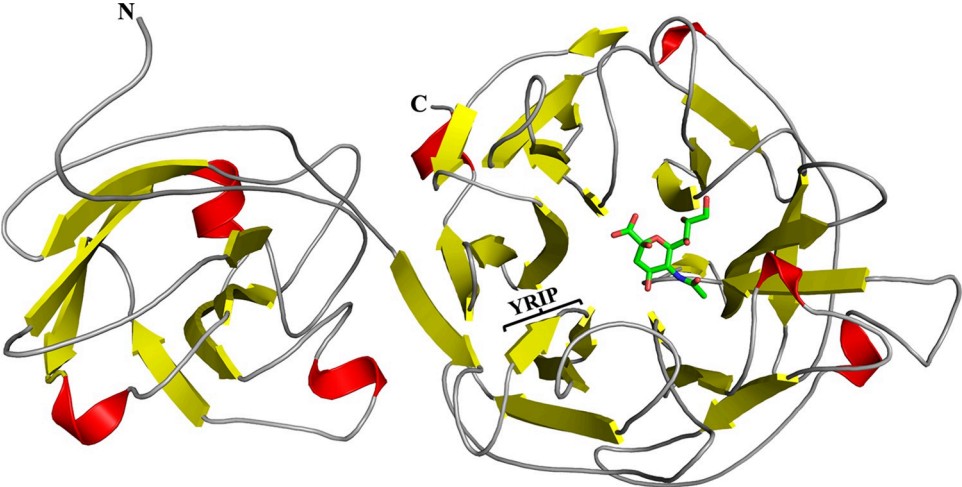

**Fig 5. Overall Architecture of PG0352.** Cartoon representation of the overall structure of PG0352 highlighting the N-terminal CBM domain and the C-terminal Sialidase domain. Helix and β-sheet secondary structures are colored red and yellow, respectively. N-Acetylneuraminic acid (Neu5Ac) is shown bound within the center of the sialidase domain, with carbon, nitrogen, and oxygen atoms colored green, blue and red, respectively. The N- and C-termini are labeled accordingly. The region labeled as "YRIP" corresponds to the conserved "F/YRIP" sequence mutated to Ala-Ala-Ala-Ala to generate the catalytically inactive PG0352 (PG0352[AAAA]).

**Table 2. Data collection and refinement statistics for six PG0352 crystal structures.**

| Crystallographic Parameter | APO | DANA | Neu5Ac | 3'-SL | 6'-SL | Fructose |
|---|---|---|---|---|---|---|
| PDB ID | | | | | | |
| Space group | P2$_1$ | P2$_1$ | P2$_1$ | P2$_1$ | P2$_1$ | P2$_1$ |
| Unit cell length (Å) a | 61.63 | 80.52 | 61.25 | 61.48 | 61.72 | 80.42 |
| b | 56.82 | 55.90 | 56.78 | 56.83 | 56.79 | 55.50 |
| c | 79.98 | 84.48 | 80.38 | 80.03 | 80.14 | 85.17 |
| β (°) | 96.13 | 115.49 | 96.07 | 96.30 | 96.33 | 115.09 |
| Resolution (Å) | 46.23–1.84 | 44.0–2.14 | 39.96–1.93 | 41.62–1.42 | 39.82–1.41 | 69.73–2.29 |
| Highest res. shell (Å) | 1.91–1.84 | 2.22–2.14 | 2.00–1.93 | 1.47–1.42 | 1.46–1.41 | 2.37–2.29 |
| Total observations [a] | 181029 (18169) | 144662 (14673) | 121443 (11319) | 455404 (44577) | 311084 (30132) | 142549 (14098) |
| Total unique | 47211 (4717) | 37392 (3718) | 40836 (3885) | 103256 (10239) | 103559 (10087) | 30671 (3060) |
| Multiplicity | 3.8 (3.9) | 3.9 (3.9) | 3.0 (2.9) | 4.4 (4.4) | 3.0 (3.0) | 4.6 (4.6) |
| Completeness (%) | 98.42 (98.50) | 98.76 (98.30) | 98.18 (94.43) | 99.78 (98.65) | 97.56 (95.23) | 98.91 (98.73) |
| I/σ(I) | 10.67 (3.52) | 7.33 (0.80) | 7.52 (0.94) | 8.66 (0.85) | 11.96 (1.04) | 7.74 (1.32) |
| R$_{merge}$ | 8.27 (35.57) | 14.58 (90.40) | 12.88 (79.13) | 11.83 (96.58) | 7.69 (98.88) | 16.01 (86.28) |
| CC$^{1/2}$ | 0.996 (0.890) | 0.991 (0.752) | 0.986 (0.556) | 0.994 (0.648) | 0.997 (0.372) | 0.988 (0.573) |
| CC* | 0.999 (0.970) | 0.998 (0.927) | 0.996 (0.845) | 0.998 (0.887) | 0.999 (0.737) | 0.997 (0.854) |
| Wilson B-Factor (Å$^2$) | 13.88 | 36.58 | 26.88 | 15.60 | 16.26 | 37.28 |
| No. non-hydrogen atoms | 4572 | 4156 | 4407 | 4828 | 4778 | 4155 |
| R$_{work}$ | 15.56 (21.46) | 18.56 (29.21) | 17.36 (28.18) | 16.35 (29.50) | 15.62 (27.90) | 17.37 (21.83) |
| R$_{free}$ | 18.88 (24.64) | 22.72 (34.56) | 21.67 (31.38) | 18.83 (32.53) | 18.01 (28.13) | 22.00 (28.54) |
| Ave. B factor, protein (Å$^2$) | 18.44 | 41.63 | 32.57 | 20.38 | 21.16 | 38.11 |
| Ave. B factor, ligand (Å$^2$) | 34.85 | 43.89 | 40.80 | 33.25 | 38.06 | 51.62 |
| Ave. B factor, solvent (Å$^2$) | 28.04 | 43.06 | 35.90 | 31.46 | 32.18 | 39.07 |
| RMSD$^c$ bond length (Å) | 0.008 | 0.007 | 0.008 | 0.009 | 0.015 | 0.007 |
| RMSD bond angle (°) | 0.920 | 0.910 | 0.990 | 1.110 | 1.460 | 0.960 |
| Ramachandran Plot | | | | | | |
| Favored (%) | 96.79 | 96.13 | 96.20 | 96.81 | 96.81 | 95.93 |
| Allowed (%) | 3.21 | 3.67 | 3.60 | 3.19 | 2.99 | 4.07 |
| Outliers (%) | 0.0 | 0.20 | 0.20 | 0.0 | 0.20 | 0.0 |

[a] Values in parentheses represent the values in the highest-resolution shell.

CBM cleft serves to bind sialoglycans for subsequent positioning of the terminal sialic acid within the active site of the sialidase domain [32,44, 46]. Despite exhibiting low sequence identity with the CBMs of Sialidase26 and *Tf*NanH, several residues are conserved within the cleft that may be involved in the coordination of substrates, including Trp-36, His-40, Lys-53, Arg-95, and Ser-169 [46–48].

Inspection of the sialidase domain of PG0352 reveals significant structural homology with other bacterial, human, and fungal members of the CAZy Glycoside Hydrolase Family 33 (GH33). A DALI search identified a sialidase from the non-pathogenic soil bacterium *Micromonospora viridifaciens* (PDB entry 1EUT) [49] as the closest structural homolog to PG0352, with a sequence identity of 32%, Z-score of 42.7, and an RMSD of 2.1Å for 333 equivalent Cα atoms. *Tf*NanH also exhibits structural homology to PG0352, with an RMSD of 1.54Å for 318 equivalent Cα atoms, despite only sharing 21% sequence identity within this domain. The residues involved in catalysis and the sequence motifs that contribute to shaping the structure of the sialidase domain for GH33 enzymes are completely conserved in PG0352. There are four Asp-box motifs located on the opposite face of the active site that serve to maintain the β-

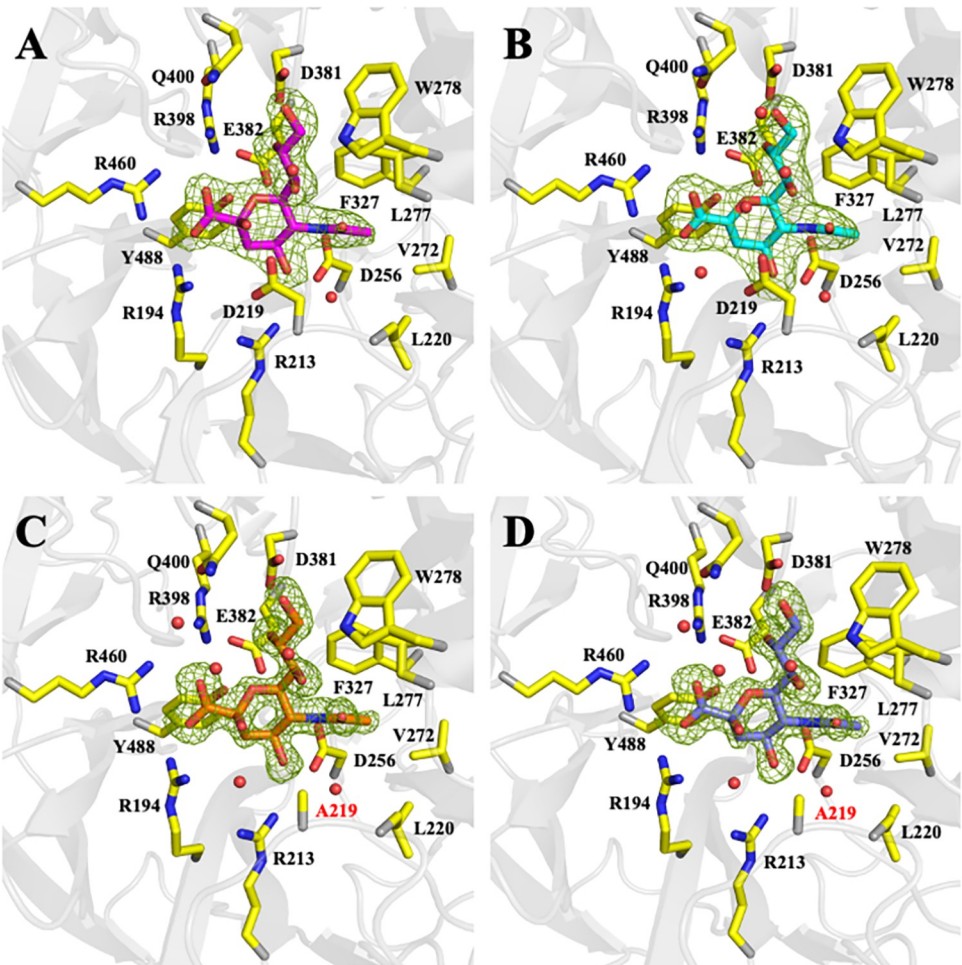

**Fig 6. PG0352 Ligand Interactions.** Cartoon representation of the sialidase active site bound with **(A)** Neu5Ac, **(B)** DANA, **(C)** 3'SL, and **(D)** 6'SL. $F_O$-$F_C$ omit electron density maps, contoured at 3s, are shown as green mesh in each panel. Residues making interactions with ligands are labeled and shown as sticks, with carbon, nitrogen, and oxygen atoms colored yellow, blue, and red, respectively. Ordered water molecules are depicted as red spheres. The D219A mutation is labeled in red in panels C and D. Coordination of the carboxylate head of sialic acid is mediated by the Tri-arg motif (Arg-194, Arg-398, Arg-460). The glycerol moiety is stabilized by interactions with Asp-381, while the N-acetyl moiety is stabilized by polar interactions with Asp-256 and insertion into a hydrophobic pocket formed by Leu-220, Val-272, Leu-277, Trp-278, and Phe-327. Chemical structures for each ligand are shown in **S6 Fig**.

propeller fold architecture of the enzyme [50]. The active site cleft is located at the center of the β-propeller and is surface accessible (**Fig 5**). As observed previously, there is a difference in the electrostatic potential on each face of the enzyme, with a more positive potential on the catalytic face and a more negative potential on the opposite face [41]. This charge differential is proposed to be crucial in orienting the catalytic face of the enzyme towards the substrate [51–53]. The positions of the side chains of Tyr-488 and Glu-382 suggest that they serve as the nucleophile and the general base catalyst, respectively, as they occupy topologically equivalent positions in other structurally characterized bacterial sialidase enzymes (**Fig 6**) [34,41,44,49,54]. Similarly, Asp-219 is positioned to function as the acid/base catalyst to activate a water molecule for release of free sialic acid [48,49,55,56].

We observed additional electron density within the active site pocket near the side chain of Tyr-488 that we subsequently modeled as a partially occupied citrate molecule derived from

the crystallization cocktail (**S4, S5E and S6** Figs). Similarly, a recent report also found a tartrate molecule at the same position [41]. One of the carboxyethyl groups of citrate interacts with Arg-194, Arg-398, and Arg-460, a triad of conserved arginine residues implicated in stabilizing the carboxylate group of sialic acid [48,49,57]. Arg-194 is part of the conserved RIP motif and is further stabilized by Glu-504 [50]. Another carboxyethyl group interacts with the side chain of Arg-213, mimicking the interaction observed between this side chain and the O4 of sialic acid [34,49,58]. This interaction perturbs the side chain of Asp-219, which exhibits alternate rotamer conformations in the unliganded structure. The third carboxyethyl group does not interact with the enzyme, given its orientation away from the active site pocket.

## PG0352 interactions with product and inhibitor

We also determined the crystal structures of PG0352 bound with the product Neu5Ac (PG0352:Neu5Ac) and the sialidase inhibitor 2-deoxy-2,3-dehydro-N-acetylneuraminic acid (PG0352:DANA) to resolutions of 1.93Å and 2.14Å, respectively (**Fig 6** and **Table 2**). The structures were solved utilizing molecular replacement and the unliganded PG0352 structure as the search model. Each structure contains one monomer of PG0352 in the asymmetric unit comprised of the CBM and sialidase domains. The calculated RMSDs between the PG0352: Neu5Ac and PG0352:DANA structures and the unliganded PG0352 structure are at 0.227Å and 0.517Å, respectively, indicating that there are no significant global differences between the structures.

Clear electron density was visible for Neu5Ac bound in its alpha anomeric form within the active site of the PG0352:Neu5Ac crystal structure (**Figs 6A and S5A**). Neu5Ac is oriented with its carboxyl group interacting with the guanidinium groups of Arg-194, Arg-398, and Arg-460. The C-2 hydroxyl forms a hydrogen bond to Asp-219, while the C-4 hydroxyl makes hydrogen bonding interactions with Asp-256, Arg-213, and Asp-219. The N-acetyl moiety sits within a hydrophobic pocket formed by Leu-220, Val-272, Leu-277, Trp-278, and Phe-327, with the N-acetyl amide interacting with the carboxylate of Asp-256. The glycerol moiety is predominantly stabilized via interactions between the O-8 and O-9 atoms and Asp-381. Collectively, the interactions position the O6 and C2 atoms of Neu5Ac at 2.89Å and 3.26Å, respectively, from the hydroxyl group of Tyr-488. The binding of Neu5Ac also results in the side chain of Asp-219 adopting a single conformation.

The RMSD between the sialidase domains of PG0352:Neu5Ac and PG0352:DANA bound crystal structures is 0.19Å, indicating that the binding of DANA does not result in significant structural deviations. As expected, DANA binds within the sialidase active site in a similar conformation to that observed for Neu5Ac, with all active site residues exhibiting nearly identical rotamer conformations (**Figs 6B and S5B**). The interactions described above involving the carboxylate, N-acetyl, and glycerol moieties of Neu5Ac are conserved with those observed for DANA bound to PG0352. DANA exhibits a more planar ring structure compared with Neu5Ac, which results in the loss of the interactions of the O6 atom with both Glu-382 and Arg-398. In addition, the hydroxyl of Tyr-488 is shifted away from the O6 atom by 0.35Å. These perturbations bring the hydroxyl group of Tyr-488 closer to the C2 atom of DANA. The planarity of the sugar ring also perturbs the positioning of the glycerol moiety, increasing the distances between the O8 and O9 atoms and the side chains of Glu-382 and Arg-398 by 0.5Å and 0.3Å, respectively. As a result, these atoms now interact with the side chain of Arg-398. Two ordered waters, absent in the PG0352:Neu5Ac crystal structure, are observed, which further provide stabilizing interactions of DANA within the active site. One of the waters forms hydrogen bonds with Arg-194 and Asp-219, likely mimicking the proposed water molecule activated by Asp-219 to facilitate the hydrolysis of the glycosidic bond [34,48,49].

To probe how PG0352 interacts with native glycans, we engineered a D219A mutation in PG0352 to inhibit the ability of the enzyme to hydrolyze the glycosidic bond [44,59]. We subsequently co-crystallized D219A PG0352 with 3'-sialyllactose (3'-SL) and 6'-sialyllactose (6'-SL) and solved their structures utilizing molecular replacement methods to 1.42Å and 1.41Å, respectively (**Fig 6** and **Table 2**). The overall architectures of the CBM and sialidase domains are completely conserved in the structures. In each structure, the sugar ring of the sialic acid moiety adopts a distorted-boat conformation, with shifts in the positions of the O2, O6, and C7 hydroxyl group of ~0.5Å when compared to DANA, due to the D219A substitution. Otherwise, the residues and interactions involved in the binding of the terminal sialic acid moiety in the PG0352:3'-SL and PG0352:6'-SL structures are essentially identical to that observed for the binding of Neu5Ac in the PG0352:Neu5Ac structure (**Figs 6C, 6D, S5C, and S5D**). Three additional ordered water molecules are observed in the active site. One ordered water occupies the space vacated by the Asp-219 side chain, coordinating the O2 atom to the main chain amide of Ala-219 and the side chain of Arg-194. The other two ordered waters are part of an interaction network that connects the carboxylate of sialic acid to the side chains of Trp-278, Arg-398, and Gln-400. No clear electron density was observed to facilitate modeling of the galactose or glucose moieties of 3'-SL or 6'-SL.

## Functional characterization of PG0352

Most bacterial sialidases are monomers despite differences in their molecular weights (MW) and domain architectures [48,60,61]. In contrast, viral sialidases typically form tetramers [60]. We used Size-Exclusion Chromatography (SEC) to determine if PG0352 forms monomers or oligomers. The SEC chromatogram showed that PG0352 was eluted at a single large peak, with an estimated MW of 52 kDa, indicating that PG0352 forms monomers in solution (**S7 Fig**). We next examined the kinetic properties of PG0352 and determined its affinity towards Neu5Ac and Neu5Gc, the two most abundant and best studied sialic acids [30,31]. For these experiments, 4-methylumbelliferyl-α-D-N-acetylneuraminic acid (4-MUNANA) and 4-methylumbelliferyl-α-D-glycolylneuraminic acid (4-MUNAGc), which represent Neu5Ac and Neu5Gc, respectively, were utilized as substrates. We first evaluated the effect of pH on the sialidase activity of PG0352 utilizing 4-MUNANA. While sialidase activity was detectable across a pH range of 3.0–8.0, maximal activity was observed at a pH 5.0 (**Fig 7A**). We then incubated PG0352 with various amounts of 4-MUNANA or 4-MUNAGc and determined the

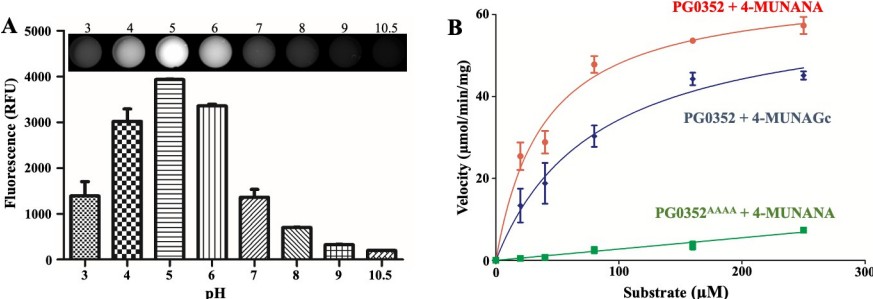

**Fig 7. Functional Evaluation of PG0352. (A)** PG0352 was treated with the fluorescence substrate 4-methylumbelliferyl-α-D-N-acetylneuraminic acid (4-MUNANA) under different pH conditions to evaluate the optimal pH for its sialidase activity as described in the Methods section. PG0352 displays a broad activity range (pH 3–8) with maximal activity at pH 5. Top: a representative image of the level of fluorescence obtained under each pH condition based on the liberation of 4-MU. **(B)** PG0352 sialidase activity was evaluated as described in the Methods section utilizing 4-MUNANA and 4-methylumbelliferyl-α-D-glycolylneuraminic acid (4-MUNAGc) as substrates. Inactive PG0352 (PG0352^AAAA) was utilized as a negative control.

initial rate of 4-methylumbelliferone (4-MU) released from each substrate. Wild-type PG0352 had a $V_{max}$ of 62.33 ± 6.236 mmol min$^{-1}$ mg$^{-1}$ and $K_m$ value of 81.65 ± 20.56 μM when 4-MUNAGc was utilized as a substrate. When 4-MUNANA was utilized as the substrate, a $V_{max}$ of 66.91 ± 3.350 mmol min$^{-1}$ mg$^{-1}$ and $K_m$ value of 39.31 ± 6.540 μM was observed. Although PG0352 exhibited a similar $V_{max}$ towards the two substrates, the $K_m$ value for 4-MUNAGc was 2-fold greater than that for 4-MUNANA, indicating PG0352 has a higher binding affinity to Neu5Ac. As expected, we did not detect any measurable sialidase activity for the PG0352$^{AAAA}$ mutant construct.

## Deletion of the CBM alters the solubility of PG0352 but the truncated protein retains sialidase activity

We sought to determine if the CBM is required for the sialidase activity of PG0352. In doing so, we removed the CBM (1–180 aa) and expressed the truncated protein (PG0352CT) that only harbors the C-terminal sialidase domain. Deletion of the CBM significantly reduced expression level of PG0352CT in *E. coli*, as well as affecting the solubility of PG0352CT, e.g., a majority of purified protein was insoluble (**Fig 8A**). After many trials, we were able to express and purify a small amount of soluble PG0352CT utilizing a pET200 vector coupled with the use of a large volume *E. coli* cultures (~ 4L) (**Fig 8A**). Although the truncated protein was not purified to homogeneity, filter paper spot assays revealed that PG0352CT still possessed sialidase activity in a time- and dose-dependent manner (**Fig 8B**). Collectively, these results indicate that the CBM is required to maintain the solubility of PG0352 but not its sialidase activity.

## The CBM binds to glycosylated serum proteins

To further investigate the role of the CBM, we expressed and purified an N-terminal truncation construct that contained only the CBM (residues 30–180). This construct was highly expressed in *E. coli*. As expected, the purified recombinant CBM protein (rCBM) did not exhibit sialidase activity. Like PG0352, the sialidase of *V. cholera* has a N-terminal lectin-like domain [48] and the sialidase of *M. viridifaciens* possesses a galactose-binding domain [46]. As

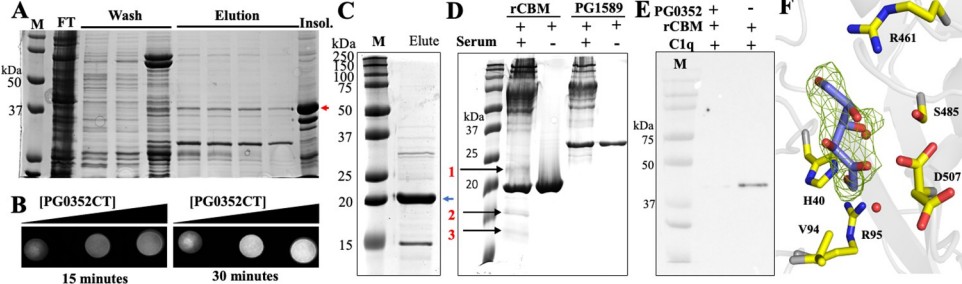

**Fig 8. Functional Evaluation of Stand-Alone CBM (carbohydrate-binding module) and Sialidase domain Constructs. (A)** SDS-PAGE gel showing the purification of the C-terminal sialidase domain construct (PG0352CT). The band pertaining to PG0352CT is indicated by the red arrow. FT: flow-through, Insol.: insoluble fraction. **(B)** Representative image from the sialidase filter spot assay of recombinant PG0352CT. **(C)** SDS-PAGE gel showing the purified recombinant CBM. **(D)** Co-immunoprecipitation (Co-IP) using rCBM and human serum. Recombinant PG1589 was included as a negative control. Red color numbers are three protein bands found in the CBM pulldown sample that are absent in the control samples. **(E)** Co-IP using rCBM and C1q untreated and treated with PG0352. For the treated sample, prior to Co-IP, C1q was incubated with PG0352 for 2 hours at 37°C. **(F)** Cartoon representation of fructose bound to the CBM domain of PG0352. $F_O$-$F_C$ omit electron density, contoured at 3σ, is shown as green mesh. Residues making interactions with fructose are labeled and shown as sticks, with carbon, nitrogen, and oxygen atoms colored yellow, blue, and red, respectively. Ordered water molecules are depicted as red spheres. The chemical structure for fructose is shown in **S6 Fig**.

these non-catalytic domains are implicated in binding carbohydrates, we speculated the same scenario may stand for the CBM. To test this hypothesis, we purified rCBM (**Fig 8C**) and conducted pull-down experiments using human serum, as it contains ample glycoproteins with various glycan modules including sialoglycans. The pull-down assay, coupled with nano LC-/MS/MS analyses, identified four proteins: C1q (25.8 kDa), immunoglobulin lambda-like polypeptide 5 (23 kDa), transthyretin (15.9 kDa), and dermcidin (12.4 kDa) (**Fig 8D**). Interestingly, these four proteins are all glycosylated [7,62,63]. To confirm this result, we repeated the pull-down assay using C1q and PG0352-treated C1q. Consistent with the above pull-down result, C1q, but not the PG0352-treated C1q, was precipitated by rCBM-resin (**Fig 8E**), suggesting that the binding between C1q and CBM is specific and most likely depends on sialylation. These results further suggest that CBM functions as a carbohydrate-binding domain.

## PG0352 CBM interactions with carbohydrates

To further characterize the mode of interaction between carbohydrates and the CBM, we co-crystallized PG0352 with fructose and determined the structure of the complex (PG0352:Fructose) to 2.29Å resolution. Superposition of PG0352:Fructose with the unliganded PG0352 structure shows an overall RMSD of 0.56Å (493 equivalent $C_\alpha$ atoms), with most of the difference localized to the CBM domain RMSD 0.527Å (145 equivalent $C_\alpha$ atoms), compared to the sialidase domain RMSD of 0.248Å (348 equivalent $C_\alpha$ atoms). The differences observed between the CBMs are due to shifts in regions distant from the fructose binding site. These shifts are unlikely to be associated with fructose binding, as they are also present in the PG0352:DANA structure, which does not contain a ligand bound within the CBM. The fructose ligand is predominantly stabilized within the CBM via direct and water-mediated hydrogen bonding interactions between its oxygen atoms and the hydrophilic side chains of His-40, Arg-95, Arg-461, Ser-485 and Asp-507, which exhibits alternate rotamer conformations (**Figs 8F and S5F**).

## Discussion

In this report, we provide biochemical and structural evidence that PG0352 confers protection against complement killing through desialylation of key complement factors such as C1q, C4 and C5. We believe that this protection primarily occurs at the gingival crevices because *Pg*, as a keystone periodontal pathogen, primarily dwells in the gingival crevices which contain a high concentration of plasma and are constantly surveilled by the complement system [64–66]. In addition to periodontal disease, *Pg* is associated with various systemic diseases such as cardiovascular disease, Alzheimer's disease, and rheumatoid arthritis [67–69]. It is conceivable that PG0352-mediated complement resistance can help facilitate the ability of *Pg* to cause systemic infections. Both basic and clinical studies indicate that sialidases play a pivotal role in the pathogenicity of *Pg* and the pathogenesis of periodontitis. First, our previous animal study showed that all the mice challenged with *Pg* W83 wild type strain developed systemic infections (e.g., secondary skin lesions and spreading infection evident in lungs, livers and other organs) and the infected mice died within 6 days after the subcutaneous injection. By contrast, the deletion mutant of PG0352 failed to induce systemic infection and the infected mice all survived [27]. Second, Moncla et al examined 25 *Pg* clinical isolates and found that they all had sialidase activity [70]. BLAST search further revealed that all the genome-sequenced *Pg* isolates contain at least one copy of PG0352 homolog. Lastly, clinical studies discovered that the level of sialidase in the gingival crevicular fluid (GCF) of periodontitis patients is significantly higher than that in healthy individuals; thus, sialidase has been proposed as a predictive biomarker to assess the severity of periodontitis and the outcome of periodontal treatment

[71,72]. In addition to *Pg*, other periodontal pathogens, such as *T. denticola* and *T. forsythia*, also produce sialidases which are important virulence factors; however, the mechanism underlying their pathogenic roles remains elusive [26,73]. Thus, the study in this report opens a new avenue to investigate the role and underlying mechanism of sialidases in other periodontal pathogens.

The oral cavity is a dynamic environment. Its pH levels can be influenced by many factors including our dietary intake. For instance, high sugar diets favor the colonization of *Streptococcus mutans* and *Lactobacilli* which can metabolize sugar (i.e., sucrose) and produce acids that demineralize teeth, causing dental caries [74]. This acid production can lower the pH levels of the dental plaque. Vroom et al demonstrated that oral biofilms can develop a pH gradient *in vitro* [75]. They prepared polymicrobial biofilms consisting of *Fusobacterium nucleatum*, *Pg*, *S. mutans*, *S. oralis*, *S. sanguinis*, and *Veillonella dispar* in a chemostat system and showed that addition of sucrose resulted in distinct pH gradients in biofilms [75]. If this is reflected *in vivo*, we could assume that the broad pH activity range for PG0352 is advantageous because it can still function despite pH fluctuations within the oral cavity. As such, the colonization and survival of *Pg* would be enhanced, given that PG0352 is critical for its pathogenicity [27,76–80].

4-MUNANA is a commonly used substrate to characterize sialidase activity [70]. This substrate consists of a fluorescent 4-MU linked to Neu5Ac. In addition to qualitative analysis, 4-MUNANA can also be used for quantitative analysis of sialidase activity [57]. Therefore, we used 4-MUNANA to kinetically characterize the sialidase activity of PG0352. We also wanted to know if PG0352 had higher affinity to Neu5Ac or Neu5Gc, the two most common forms of sialic acids. Zaramela et al reported that sialidases from gut bacteria demonstrate higher specificity towards Neu5Gc in response to dietary changes (i.e., increased red meat) [43]. As *Pg* resides in the oral cavity, it can get direct access to Neu5Gc from our diets such as red meat. To test this, we also included 4-MUNAGc to determine the affinity of PG0352 to Neu5Ac and Neu5Gc. Kinetic studies showed that PG0352 has activity towards both substrates. However, the $K_m$ for 4-MUNAGc is 2-fold higher than that for 4-MUNANA. The higher affinity towards Neu5Ac is reasonable given that it is the predominant form in the human body because humans can only produce Neu5Ac [81]. Neu5Gc can be acquired through dietary means. Having activity towards both Neu5Ac and Neu5Gc renders *Pg* ability to scavenge sialic acids from the food we eat. This could potentially contribute to oral microbiota at the community level by liberating sialic acids from many different sources. For instance, *F. nucleatum* can use sialic acids for its growth, but it does not produce sialidase. Instead, *F. nucleatum* produces a sialic acid transporter, denoted as SiaT, which allows it to acquire sialic acids liberated by sialidases from other bacteria (e.g., *Pg*) [82]. With a broad range of activity and substrates, PG0352 can function and remove sialic acids from diverse substrates under different conditions which can be utilized by *Pg* and/or other bacteria.

Our previous report uncovers that PG0352 protects *Pg* from complement killing and yet its underlying mechanism is not fully understood [27]. Herein, we provide several lines of evidence that this protection is most likely due to its desialylation activity against several key complement factors (e.g., IgG, C1q, and C4). Due to aforementioned technique limitation, however, we were unable to demonstrate that desialylation directly confers *Pg* complement resistance even though we showed this activity protects the surrogate *E. coli* strain from the serum killing. There are over 30 proteins in the complement system, including plasma proteins, regulatory factors, receptors, and ligands, some of which are modified by sialoglycans [7]. These sialoglycans contribute to protein function, stability, protein-protein interactions, and self-recognition [7,28]. For instance, sialic acid is believed to minimize non-specific interactions of the C1 complex by stabilizing C1q [7]. Desialylation of FH can alter its function and lead to pathogenic effects [29]. We demonstrate that PG0352 can efficiently desialylate both

human serum and individual complement factors, including C1q, C4, C5, FH, FI, C4bp, and IgG, highlighting its importance in complement evasion and its potential link to altering host immune homeostasis, both of which play a critical role in the pathogenesis of periodontitis and its association with systemic illnesses [2,64,69,83]. In addition to disarming key complement factors, it is also possible that PG0352 impairs the complement system through other mechanisms, e.g., modifying *Pg* surface molecules such as polysaccharide capsules and lipopolysaccharide (LPS) with sialic acids, which in turn confers it serum resistance. For instance, *N. gonorrohoeae* incorporates sialic acid into its lipopolysaccharide (LOS) to capture circulating FH for complement evasion [84,85]. A similar scenario may exist in *Pg*, given that its LPS contains a high level of sialic acids [86]. Additionally, gingipains, a group of Arg-X and Lys-X proteases, protect *Pg* from serum killing through degradation of complement factors such as C3 and C5 [66,87,88]. Aruni et al reported that deletion of PG0352 diminished the gingipain activity [76]. Thus, it is also possible that PG0352 contributes to serum resistance through controlling the activity of gingipains.

PG0352 is a dual domain protein consisting of a CBM and a canonical sialidase domain. BLASTP analysis indicates that the CBM is highly conserved in sialidases produced by different *Porphyromonas* species isolated from humans and animals [89]. Initially, we intended to determine if the CBM domain is required for sialidase activity. To address this question, we engineered a truncated construct comprised of just the C-terminal sialidase domain (PG0352CT). To our surprise, this truncated protein was largely insoluble. After many attempts, a trace amount of soluble protein was expressed. We were subsequently able to purify a small amount of PG0352CT with a detectable sialidase activity, suggesting that the CBM is not required for the sialidase activity of PG0352.

The exact function of the CBM is not fully understood. Our crystal structure data suggests it is a putative carbohydrate-binding module that belongs to the CBM93 family. Like PG0352, the sialidase of *V. cholera* has dual lectin-like domains, one of which was shown to bind sialic acids to coordinate its activity [48,90]. The NanH sialidase of *T. forsythia* also harbors a putative CBM [32]. Taken together, we reasoned that CBM might bind to sialoglycans to modulate PG0352 substrate specificity. To test this hypothesis, we performed pull-down assays using purified CBM and human serum. Our data, though preliminary, showed that several glycoproteins were precipitated by the CBM, including C1q, a heavily glycosylated complement factor [7]. In addition, our crystal structure data also support that the CBM functions to bind carbohydrates. To substantiate these results and determine the types of carbohydrates the CBM binds to, we conducted high throughput glycan arrays in two independent entities (ZBiotech and the National Center for Functional Glycomics; NCFG). The glycan array from ZBiotech, which contains 100 different glycans, showed that the CBM bound to several candidate glycans; however, the binding activity was too weak to draw definitive conclusions. The NCFG glycan array, which contains 562 glycans, showed that the CBM exhibits low binding to several large polylactosamines.

There is no clear electron density within the PG0352:3'-SL or PG0352:6'-SL crystal structures to facilitate the modeling of the galactose or glucose moieties. The absence of electron density for these moieties suggests that there are no specific interactions between PG0352 and glycan substrates at the active site, except for the terminal sialic acid. While it is possible that 3'-SL and 6'-SL have undergone hydrolysis during co-crystallization, it is unlikely, as stability studies of 6'-SL indicate, that the molecule remains stable for up to one month in a neutral pH solution at 35˚C [91]. Recent studies have highlighted the importance of CH-π stacking interactions to carbohydrate binding in enzymes, particularly glycan-modifying enzymes like sialidases [92–94]. These studies have highlighted interactions between aromatic residues (i.e., Trp, Tyr, Phe, and His) and non-covalently linked carbohydrates among PDB deposited

glycan-containing structures, imparting significant, non-specific binding affinity to carbohydrates (~50% and 90% of binding energy and affinity, respectively) [93]. Additionally, the loops present above the active site, particularly above the carboxylate-binding region, have been implicated in substrate selectivity among sialidases [41,44,47,49,52,57,95]. These loops have previously been suggested to impart substrate selectivity via steric hindrance mediated by aromatic residues. However, *Trypanosoma cruzi* sialidase (PDB ID 1MS0) shows a CH-π stacking interaction with a co-crystallized lactose molecule [96]. Interestingly, PG0352 does not display a linkage preference [77], nor does it possess aromatic residues within this region, while many other sialidases with linkage preferences do [34,44,96]. Thus, the broad substrate preference of PG0352 may be mediated by the lack of specific aromatic residues within the loops above the active site and, more broadly, linkage preference within sialidases may be mediated by aromatic residue mediated CH-π stacking.

During the preparation of this manuscript, the structure of PG0352 bound to tartrate was published (41). The structure was determined to 2.10Å in space group $P3_12_1$, with 3 molecules in the asymmetric unit. Superposition of the structures reported here with this structure indicates that there are no significant differences in the overall architecture or within the N- and C-terminal domains. Interestingly, the tartrate molecule observed in the sialidase active site of their structure binds in a virtually identical conformation to that observed for citrate bound in the unliganded PG0352 structure reported here, suggesting that these molecules may serve as useful stabilizing ligands to facilitate the crystallization of sialidase enzymes.

## Materials and methods

### Preparation of PG0352 recombinant proteins

The nucleotide sequence encoding PG0352 was codon optimized (GenScript), synthesized, and then inserted into a pUC57 cloning vector. The codon-optimized amplicon was released by BamHI and HindIII and subcloned into the pQE80L expression vector (Qiagen, Germantown, MD). The resulting expression vector was then transformed into BL21-Star (DE3) for protein expression and purification. To prepare PG0352CT protein, the sequence encoding the C-terminal sialidase domain (residues 180–526) was PCR amplified, cloned into pET200, verified by DNA sequencing, and then transformed into BL21-Star (DE3). To prepare recombinant CBM, the nucleotide sequence encoding the N-terminal CBM of PG0352 (residues 30–180) was PCR amplified, cloned into pET101, verified by DNA sequencing, and subsequently transformed into BL21-Star (DE3) for expression. The PCR primers for cloning can be found be found in S1 Table.

The expression of recombinant proteins in BL21-Star (DE3) was induced with 1 mM isopropyl-β-D-thiogalactopyranoside (IPTG) at 30°C for WT PG0352 and 16°C for PG0352CT and CBM. Recombinant proteins were purified under native conditions as described previously [26]. In brief, *E. coli* cell pellets were resuspended in lysis buffer (300 mM NaCl, 50 mM NaH$_2$PO$_4$, and 10 mM Imidazole, pH 8.0) supplemented with lysozyme (10 mg/mL) and benzonase (0.5 µg/mL) and lysed using an Emulsiflex-C3 (Avestin; Ontario, Canada). After centrifugation at 15,000 RPM for 20 minutes at 4°C, supernatants were collected and applied to a HisTrap HP Ni-NTA column (Cytiva, Marlborough, MA) using the NGC FPLC System (Bio-Rad, Hercules, CA). Size Exclusion Chromatography (SEC) was conducted to measure the size of PG0352 and to improve its purity. Ni-NTA purified PG0352 proteins were pooled and dialyzed overnight in a dialysis buffer (50 mM Tris, 150 mM NaCl, pH 7.0). The dialyzed proteins were applied to an XK 16/100 column packed with Superdex 200 resin (Cytiva, Marlborough, MA) or a Superdex 200 Increase 10/300 GL (Cytiva).

## Site-directed mutagenesis to produce catalytically inactive PG0352

Site-directed mutagenesis was used to create a catalytically inactive PG0352 protein. A construct in which Tyr-193, Arg-194, Ile-195 and Pro-196 were collectively mutated to alanine (PG0352[AAAA]) was engineered using a Q5 site-directed mutagenesis kit (New England Biolabs, Ipswich, MA), as previously reported [32]. Forward primers were designed to contain the point mutations and reverse primers were designed with an adapter (overhang) to the forward primer. The wild-type codon optimized PG0352 gene cloned in pQE80 was used as a template for site-directed mutagenesis using PCR amplifications. The subsequent PCR products were treated with a proprietary mix of kinase, ligase, and DpnI enzymes (New England Biolabs), and then transformed into NEB5α for plasmid purification. The purified plasmids were verified by DNA sequencing and subsequently transformed into BL21-Star (DE3) for protein expression and purification. The D219A mutation utilized for crystallographic structure determination was generated utilizing SuperFi II DNA Polymerase coupled with whole-plasmid PCR-based methods and the pQE80L plasmid containing an N-terminal His$_6$ affinity tag and residues 31–526 of PG0352 as the template. PCR product was treated for 1 hour at 37°C with 1x DpnI (New England Biolabs) before transformation into *E. coli* NEB5α cells. The mutation was verified using whole-plasmid sequencing. The primers for the site-directed mutagenesis can be found in **S1 Table**.

## Sialidase activity assays

We measured the sialidase activity of PG0352 by carrying out the assay under different pH conditions. This assay was performed on black 96-well polystyrene plates designed for measuring fluorescence (Corning Inc., Corning, NY). 5 nM PG0352 was used to treat 0.1 mM 4-methylumbelliferyl-α-D-N-acetylneuraminic acid (4-MUNANA) for 1 minute at 37 °C in: 20 mM sodium citrate (pH 3–5), 20 mM sodium phosphate (pH 6–8), and 20 mM sodium bicarbonate (pH 8–10.5) [26,32]. Following incubation, the reactions were quenched with sodium bicarbonate (pH 10.5) at a volume ratio of 1:1.5 (reaction:buffer). Sialidase activity was quantified by measuring 4-methylumbelliferone (4-MU; Sigma-Aldrich, St. Louis, MO) fluorescence ($\lambda_{ex}$ = 350 nm and $\lambda_{em}$ = 450 nm) on a Varioskan LUX Multimode Microplate Reader (Thermo Fisher, Pittsburgh, PA). For kinetic characterization, 5 nM of PG0352 was incubated with 0–250 μM 4-MUNANA in a reaction buffer containing 10 mM sodium citrate, pH 5.0. Reactions were carried out for 1 minute at 37 °C and immediately quenched with 10 mM sodium bicarbonate, pH 10.5 at a volume ratio of 1:1.5. A standard curve was generated using 4-MU at a set of concentrations (0–250 nM). PG0352 activity was applied against the standard curve to measure the amount of 4-MU released ($\mu$mol min$^{-1}$ mg$^{-1}$). Data were fit to Michaelis-Menten kinetics using GraphPad Prism 5 (Graph-Pad Software, San Diego, CA) to calculate $V_{max}$ and $K_m$. To evaluate substrate specificity for PG0352, the same kinetic studies were performed using 4-methylumbelliferyl-α-D-glycolylneuraminic acid (4-MUNAGc; Sussex Research, Ottawa, Canada). Sialidase from *Clostridium perfringens* was included as a control for all conditions.

## Serum killing protection assay with PG0352

For this assay, normal human serum (NHS) was pre-treated with wild-type PG0352 or PG0352[AAAA] to determine if PG0352 can inhibit serum bactericidal activity against *E. coli*, as previously described with some modifications [26,27]. Briefly, *E. coli* NEB5α was cultivated in LB media until it reached the exponential growth phase (OD$_{600}$ 0.5–0.7). This strain was chosen due to its sensitivity to serum killing [40]. The cells were spun down and washed twice with gelatin barbiturate veronal buffer (GVB$^{++}$)[40]. During this time, NHS was thawed and

diluted in GVB$^{++}$ with 10 mM sodium citrate, pH 5.0 to final concentrations of either 2% (v/v) or 5% (v/v). The serum was treated with 100 nM wild-type PG0352 or PG0352$^{AAAA}$ for 30 minutes at 37˚C. The treated serum samples were co-incubated with NEB5α cells at 37˚C for 30 or 60 minutes. After the incubation, the samples were serially diluted and plated on to LB agar plates for colony counting. Plates were incubated at 37˚C overnight. The next morning, the colonies on the plates were enumerated to calculate survival rates. Survival rates were calculated as follows: numbers of colonies from treated serum in relative to those from heat-inactivated human serum.

### Complement deposition assays

This experiment was conducted as previously reported with some modifications [26]. In brief, *E. coli* NEB5α cells were grown to mid-log (OD$_{600}$: 0.5–0.7), harvested by centrifugations, washed, and resuspended with GVB$^{++}$. To measure complement deposition, $10^7$ NEB5α cells were co-incubated with either 5% normal human serum or PG0352-treated serum (co-incubated with 100 nM recombinant PG0352 for 30 mins at 37˚C) for 20 mins at 37˚C. After the incubation, the resulted *E. coli* cells were incubated on ice for 1 min to inactivate complement activation harvested by centrifugation. The obtained cell pellets were washed with ice-cold PBS-EDTA and then subjected to SDS-PAGE, followed by immunoblotting analysis using antibodies C3 (cat#, ab48611), C5b-9 (cat#, ab66768), and C9 (cat#, ab17931) or *E. coli* GroEL (cat#, ab82592, loading control). These antibodies were purchased from Abcam.

### Treatment of human serum and complement factors with PG0352

As described previously [26], human serum at a final concentration of 0.15% (v/v) was treated with 0.2–0.6 μg of either wild-type PG0352 or PG0352$^{AAAA}$ in 10 mM sodium citrate, pH 5.0. The samples were incubated at 37˚C and samples were taken at different time points (e.g., 15, 30, and 60 minutes). The samples were subjected to SDS-PAGE, followed by lectin blot analysis using Sambucus nigra (SNA; Vector Laboratories, Newark, CA) and Concanavalin A (ConA; Vector Laboratories) lectins. Similar experiments were performed using the human complement factors, including C1q, C3, C4, C5, Factor H (FH), Factor I (FI), and C4 binding protein (C4bp). Per the manufacturer (Complement Technologies, Tyler TX) website, these factors were purified from pooled serum from healthy donors. Human immunoglobin IgG was purchased from Sigma-Aldrich. The complement proteins were treated with 100 nM of wild-type PG0352, PG0352$^{AAAA}$, or *C. perfringens* sialidase (control). After the treatment, samples were collected at 30 and 60 minutes, subjected to SDS-PAGE, and analyzed utilizing lectin blot analyses.

### Lectin blot analysis

Lectin blots were performed as previously described [26,27]. Briefly, serum samples or complement proteins were subjected to SDS-PAGE and then transferred to PVDF membranes. The resultant membranes were blocked overnight in 1x carbo-free blocking buffer (Vector Laboratories) supplemented with 0.05% Tween-20. Membranes were then incubated with 0.2 μg/mL SNA or 0.5 μg/mL ConA in 0.2x carbo-free blocking buffer with 0.05% Tween-20 for 1 hour at room temperature. Following incubation, membranes were washed four times with filtered PBS, 0.05% Tween-20 (PBS-T), and then incubated with streptavidin-horseradish peroxidase conjugate (Thermo Fisher) for 1 hour. Blots were subsequently washed four times with filtered PBS-T, developed using an ECL luminol assay (Bio-Rad), and imaged using a ChemiDoc Imaging System (Bio-Rad).

## In-gel trypsin digestion

Protein gel samples were cut into ~1 mm cubes, digested with trypsin, followed by extraction of tryptic peptides. The excised gel pieces were washed consecutively in 200 μL distilled/deionized water, 200 μL 50 mM ammonium bicarbonate, 50% acetonitrile, and finally 200 μL of 100% acetonitrile. The dehydrated gel pieces were reduced with 30 μL 10 mM dithiothreitol in 100 mM ammonium bicarbonate for 1 hour at 56˚C and alkylated with 30 μL 55 mM iodoacetamide in 100 mM ammonium bicarbonate at room temperature in the dark for 45 minutes. Wash steps were repeated as described above. The gel was then dried and rehydrated with trypsin at an estimated ratio of 1:10 (w/w) in 50 mM ammonium bicarbonate, 10% acetonitrile and incubated at 37˚C for 18 hours. Digested peptides were extracted twice with 200 μL 50% acetonitrile, 5% formic acid, and once with 100 μL 75% acetonitrile, 5% formic acid. Extractions from each sample were pooled together, filtered with a Costar Spin-X 0.22 μm spin filter (Corning Inc., Corning, NY) and dried in a speed vacuum. Each sample was reconstituted into 0.5% formic acid prior to LC-MS/MS analysis.

## Nano LC-ESI-MS/MS analysis

Purified FH was treated with wild-type PG0352 for 3 hours and subjected to SDS-PAGE with Coomassie blue staining. Two molecular weight bands of interest were gel extracted from treated and untreated FH samples and submitted to the Proteomics and Metabolomics Facility at the Cornell Institute of Biotechnology for N-link glycosylation analysis using nanoscale liquid chromatography coupled to tandem mass spectrometry (nano LC-ESI/MS/MS). The samples were treated with trypsin and reconstituted in 0.5% formic acid prior to nano-LC-ESI-MS/MS analysis, which was carried out using an Orbitrap Fusion Tribrid mass spectrometer equipped with a nano-spray Flex Ion Source, coupled with a Dionex UltiMate 3000 RSLCnano system (Thermo, Sunnyvale, CA). 5 μL peptide samples were injected onto a Pep-Map C-18 RP nano trapping column (5 μm, 100 μm i.d x 20 mm) at a 20 μL/min flow rate for rapid sample loading and then separated on a PepMap C-18 RP nano column (2 μm, 75 μm x 25 cm) at 35˚C. Tryptic peptides were eluted using a 60-min gradient of 5% to 35% acetonitrile in 0.1% formic acid at 300 nL/min, followed by an 8-min ramping to 90% acetonitrile, and an 8-min hold at 90% acetonitrile. The column was re-equilibrated with 0.1% formic acid for 25 minutes prior to the next run. The Orbitrap Fusion was operated in positive ion mode with spray voltage set at 1.5 kV and source temperature at 275˚C. External calibration for Fourier transform (FT), Ion Trap (IT) and quadrupole mass analyzers were performed. In data-dependent acquisition (DDA) analysis, the instrument was operated using FT mass analyzer in MS scan to select precursor ions followed by second "Top Speed" data-dependent Higher energy C-trap dissociation (HCD) and Electron-transfer/Higher Energy collision dissociation (EThcD) toggle method. The HCD in orbitrap MS/MS scans at 3 m/z quadrupole isolation were for precursor peptides with 2–3 charged ions above a threshold ion count of 10,000 and normalized collision energy of 30% MS survey scans at a resolving power of 120,000 (FWHM at m/z 200), for the mass range of m/z 350–1600. Dynamic exclusion parameters were set at 35 s of exclusion duration with ±10 ppm exclusion mass width. Meanwhile, the EThcD in ion trap with normalized automatic gain control of 3e4 and maximum injection time of 118 ms were used for ions with charges of 3–7. All data were acquired under Xcalibur 4.4 operation software (Thermo Fisher Scientific).

## Data analysis of nano LC-ESI-MS/MS

MS and MS/MS raw spectra from each sample were searched using Byonics v. 3.6.8 (Protein Metrics, San Carlos, CA) using *Homo sapiens* Uniprot protein database. The peptide search

parameters were as follows: two missed cleavages for full trypsin digestion with fixed carbamidomethyl modification of cysteine, variable modifications of methionine oxidation and deamidation on asparagine/ glutamine residues. The peptide mass tolerance was 10 ppm and fragment mass tolerance values for HCD and EThcD spectra were 0.05Da and 0.6Da, respectively. The maximum number of common and rare modifications were set at two. The glycan search was performed against a list of 309 mammalian N-linked glycans. Identified peptides were filtered for maximum 1% false discovery rate.

## Pull-down assay

Recombinant CBM protein (rCBM) was purified using a HisTrap HP Ni-NTA column, followed by dialysis overnight against 300 mM NaCl, 50 mM $NaH_2PO_4$, and 10 mM imidazole, pH 8.0 to remove excess imidazole. The dialyzed protein was rebound to fresh Ni-NTA agarose equilibrated in the same buffer to produce rCBM-bound Ni-NTA resin (rCBM-resin). Human serum was diluted to 50% (v/v) with sterile PBS, followed by incubation with rCBM-resin overnight at 4°C with inversion. The rCBM-resin was loaded into a gravity flow column and washed with 300 mM NaCl, 50 mM $NaH_2PO_4$,10 mM imidazole, pH 8.0. After washing, the rCBM-resin was resuspended in wash buffer and boiled for SDS-PAGE analysis. In parallel, purified PG1589, a dihydropteroate synthase, was treated in an identical manner to serve as a negative control. We also included buffer-equilibrated Ni-NTA resin alone as an additional control. These controls were used to exclude nonspecific binding of serum proteins to resin. Proteins that were specifically pulled down by rCBM were excised and submitted to the Proteomics and Metabolomics Facility at the Cornell Institute of Biotechnology for protein identification using nano-LC-MS/MS as described above.

The DDA raw files with MS and MS/MS were subjected to database searches using Proteome Discoverer (PD) 2.4 software (Thermo Fisher Scientific) with the Sequest HT algorithm. The PD 2.4 processing workflow containing an additional node of Minora Feature Detector for precursor ion-based quantification was used for protein identification and relative quantitation of identified peptides and their modified forms. The database search was conducted against the *Homo sapiens* NCBI database. The peptide precursor tolerance was set to 10 ppm and fragment ion tolerance was set to 0.6 Da. Oxidation of M and deamidation of N and Q were specified as dynamic modifications of amino acid residues; protein N-terminal acetylation, M-loss and M-loss plus acetylation were set as a variable modification; carbamidomethyl C was specified as a static modification. Only high confidence peptides defined by Sequest HT with a 1% FDR by Percolator were considered for confident peptide identification.

## Crystallography

Wild-type and D219A PG0352 were transformed into *E. coli* BL21 Star (DE3). For large-scale expression, a 1 L shaker flask containing LB media was inoculated with 25 mL of starter culture at 37°C and grown to an $OD_{600}$ 0.4–0.6. The cells were induced by the addition of IPTG to a final concentration of 1 mM and the temperature was lowered to 18°C. Cells were harvested 16 hours post-induction via centrifugation and frozen at -80°C until further use. The cell pellet corresponding to 1L of cell growth medium was resuspended in Buffer A (50 mM Sodium Phosphate, pH 8.0, 300 mM NaCl, 10 mM imidazole, 0.1% Tween-20 (v/v), 0.5 mg/mL lysozyme, and 60 U/mL benzonase) and lysed using a sonicator. The cell lysate was clarified by centrifugation at 40,000*g* for 20 minutes at 4°C. The resulting supernatant was incubated with 5 mL Cobalt-NTA resin (Thermo Fisher, Rockford, IL) pre-equilibrated in buffer A for 30 minutes at 4°C with gentle mixing. The resin was poured into a column support and washed with 10 column volumes (CV) of Buffer B (50 mM Sodium Phosphate, pH 8.0, 300 mM NaCl,

and 20 mM imidazole). The protein was eluted from the column using 5 CV of Buffer C (Buffer B + 300 mM imidazole). Elution fractions were pooled and concentrated to 2 mL using an Ultra Centrifugal Filter with a 30 kDa cutoff (Millipore, Bedford, MA). The concentrated sample was applied to a HiLoad 16/600 Superdex 200 PG size-exclusion column (Cytiva, Marlborough, MA) equilibrated in 20 mM Tris, pH 8.0, and 150 mM NaCl. The peak corresponding to PG0352 was pooled and concentrated to 25 mg/mL using an Ultra Centrifugal Filter with a 30kDa cutoff.

Initial crystallization screening was carried out utilizing commercial screening kits and the sitting-drop vapor diffusion method, and identified a lead from condition H5 of the Morpheus Screen (Molecular Dimensions, Maumee, OH). These crystals appeared as urchins and could not be optimized. We subsequently harvested the crystals and generated seed stocks that were utilized in conjunction with additional screening experiments. Unliganded PG0352 crystals suitable for diffraction experiments were grown at 23°C in sitting drops by combining 3 µL protein solution at 7 mg/mL with 2 µL 1.6–1.9 M ammonium citrate tribasic, pH 7.0, and 1µL of a 1:1000 dilution of seed stock and equilibrating over a 1000 mL reservoir solution of 1.8 M ammonium citrate tribasic, pH 7.0. Complexes of PG0352 with 3'-sialyllactose (3'-SL), 6'-sialyllactose (6'-SL) and fructose were generated via co-crystallization utilizing the conditions described above, with the addition of 3'-SL, 6'-SL and fructose to the drop at final concentrations of 5 mM. To generate the N-acetylneuraminic acid (Neu5Ac) and 2-deoxy-2,3-dehydro-N-acetyl neuraminic acid (DANA) complexes, unliganded PG0352 crystals were soaked in 1.8 M ammonium citrate tribasic, pH 7.0, and 0.1M HEPES, pH 7.5 containing either 27 mM Neu5Ac or 10 mM DANA, for 60 minutes and 20 minutes, respectively. All crystals were subsequently harvested and plunge-frozen in liquid nitrogen for diffraction analysis.

X-ray diffraction data were collected at 100K on beamlines 23-ID-B (unliganded PG0352, PG0352:DANA and PG0352:fructose) and 23-ID-D (PG0352:Neu5Ac, PG0352:3'-SL, and PG0352:6'-SL) at the Advanced Photon Source (Argonne National Laboratory) utilizing Dectris Eiger-16M and Pilatus3 6M detectors, respectively. Data were processed with *Xia2/DIALS* [97,98] in the CCP4 (v8.0) suite of programs [99]. The structure of unliganded PG0352 was determined by molecular replacement (MR) using *PHASER* [100] and search models derived from a predicted structure of PG0352 (UniProt ID: Q7MX62) were generated by AlphaFold2 [101] downloaded from the AlphaFold DB [102]. Two search models were generated from the original AlphaFold model, one encompassing the N-terminal domain and one comprising the sialidase domain. Each individual domain was located during the search, with the asymmetric unit containing one complete PG0352 protein. The structure was initially refined employing a simulated annealing protocol in *phenix.refine* [103]. Successive rounds of manual model building in *COOT* [104], and refinement were used to build in missing pieces of the protein. Subsequent structures bound with Neu5Ac, DANA, 3'-SL, 6'-SL, and fructose were determined utilizing MR and the unliganded PG0352 structure as the search model, followed by employment of iterative rounds of model building and refinement as outlined above. In the final rounds of refinement, waters and ligands were added and Translation-Libration-Screw (TLS) refinement [105] was applied. Data collection and refinement statistics for all six structures are summarized in **Table 2**. Structure validation was performed with *MolProbity* [106]. Coordinates and structure factors have been deposited in the Protein Data Bank: PG0352, entry 8FEB; PG0352:Neu5Ac, entry 8T1Z; PG0352:DANA, entry 8T1Y; PG0352:3'-SL, entry 8T26; PG0352:6'-SL, entry 8T27; and PG0352:fructose, entry 8T24. LIGPLOTS were generated using *LIGPLOT+* (v.2.2.8)[107].

## Supporting information

**S1 Table. Primers used in this study.**
(PDF)

**S2 Table. MS/MS spectra for nano-LC/MS/MS for human C1q before and after treatment of PG0352.**
(PDF)

**S3 Table. MS/MS spectra for nano-LC/MS/MS for human C1q before and after treatment of PG0352.**
(PDF)

**S1 Fig. SDS-PAGE analysis of complement factors (cFactors) treated with PG0352.** For this experiment, individual complement factors were treated with wild-type PG0352 or inactive PG0352 (PG0352^AAAA) for 60 minutes and then subjected to 10% SDS-PAGE followed by Coomassie blue staining. FH: factor H; FI: factor I.
(TIFF)

**S2 Fig. PG0352 desialylates C1q.** A representative LC-ESI-MS/MS spectrum of C1q glyco-peptide untreated (top) and treated (lower) with PG0352. Underlined (red) are glycans with terminal Neu5Ac detected.
(TIFF)

**S3 Fig. PG0352 desialylates C1q.** A representative LC-ESI-MS/MS spectrum of C1q glyco-peptide untreated (top) and treated (lower) with PG0352.
(TIFF)

**S4 Fig. PG0352 Interactions with Citrate.** Cartoon representation of the sialidase active site bound with citrate. Residues belonging to the active site are labeled and shown as sticks, with carbon, nitrogen, and oxygen atoms colored salmon, blue, and red, respectively. Ordered water molecules are depicted as red spheres. Note Gln-400, Asp-219, and Asp-256 display alternate rotameric conformations. Interactions between one carboxyethyl head and the tri-Arg (Arg-194, Arg-398, Arg-460) motif are observed, proposed to mimic the interaction of the carboxylate head of sialic acid. Additional stabilizing interactions are observed between a second carboxyethyl group and Arg-213, mimicking the interaction between this side chain and the O-4 of sialic acid. The third carboxyethyl group is oriented away from the active site and does not directly interact with the enzyme.
(TIFF)

**S5 Fig. *LIGPLOT*s Corresponding to PG0352 Ligands.** Depicted are *LIGPLOT*s highlighting the interactions between PG0352 and **(A)** Neu5Ac, **(B)** DANA, **(C)** 3'SL, **(D)** 6'SL, **(E)** Citrate, **(F)** Fructose. Note **(C)** and **(D)** were crystallized with a PG0352 D219A mutant. Note water molecules were excluded from the *LIGPLOT* analysis.
(TIFF)

**S6 Fig. Chemical Structures for Ligands Used in this Study.** Note for PG0352:3'SL and PG0352:6'SL only density corresponding to the sialic acid (Neu5Ac) moiety was observed (i.e., no density was observed for the galactose or glucose moieties).
(TIFF)

**S7 Fig. PG0352 is monomeric.** The oligomeric state of PG0352 was determined using size-exclusion chromatography (SEC). The estimated molecular weight of proteins under each

peak were determined from the standard curve as described in the methods.
(TIFF)

## Acknowledgments

We thank the Proteomics and Metabolomics Facility of Cornell University for providing the mass spectrometry data and NIH SIG grant 1S10 OD017992-01 support for the Orbitrap Fusion mass spectrometer.

## Author Contributions

**Conceptualization:** Michael G. Malkowski, Chunhao Li.

**Data curation:** Nicholas D. Clark, Christopher Pham, Kurni Kurniyati, Qin Fu, Sheng Zhang, Michael G. Malkowski, Chunhao Li.

**Formal analysis:** Qin Fu, Sheng Zhang, Michael G. Malkowski, Chunhao Li.

**Funding acquisition:** Michael G. Malkowski, Chunhao Li.

**Investigation:** Nicholas D. Clark, Christopher Pham, Kurni Kurniyati, Qin Fu, Sheng Zhang, Michael G. Malkowski, Chunhao Li.

**Methodology:** Nicholas D. Clark, Christopher Pham, Kurni Kurniyati, Ching Wooen Sze, Laurynn Coleman, Qin Fu, Sheng Zhang, Michael G. Malkowski, Chunhao Li.

**Project administration:** Michael G. Malkowski, Chunhao Li.

**Resources:** Chunhao Li.

**Supervision:** Sheng Zhang, Michael G. Malkowski, Chunhao Li.

**Validation:** Christopher Pham, Kurni Kurniyati, Qin Fu, Michael G. Malkowski, Chunhao Li.

**Visualization:** Nicholas D. Clark, Christopher Pham, Kurni Kurniyati, Qin Fu, Michael G. Malkowski, Chunhao Li.

**Writing – original draft:** Nicholas D. Clark, Christopher Pham, Michael G. Malkowski, Chunhao Li.

**Writing – review & editing:** Nicholas D. Clark, Christopher Pham, Sheng Zhang, Michael G. Malkowski, Chunhao Li.

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
