## [Decision Letter · Decision Letter 0]

10 Aug 2023

Dear Professor Li,

Thank you very much for submitting your manuscript "Functional and structural analyses reveal that a dual domain sialidase protects bacteria from complement killing through desialylation of complement factors" for consideration at PLOS Pathogens. As with all papers reviewed by the journal, your manuscript was reviewed by members of the editorial board and by several independent reviewers. In light of the reviews (below this email), we would like to invite the resubmission of a significantly-revised version that takes into account the reviewers' comments.

While you will see that there was significant support for your report, there were a few issues that require your input, particularly the Major Issues cited by Reviewer 2. 

We cannot make any decision about publication until we have seen the revised manuscript and your response to the reviewers' comments. Your revised manuscript is also likely to be sent to reviewers for further evaluation.

Sincerely,

Jon T. Skare

Guest Editor

PLOS Pathogens

Karla Satchell

Section Editor

PLOS Pathogens

Kasturi Haldar

Editor-in-Chief

PLOS Pathogens

orcid.org/0000-0001-5065-158X

Michael Malim

Editor-in-Chief

PLOS Pathogens

orcid.org/0000-0002-7699-2064

Reviewer's Responses to Questions

**Part I - Summary**

Reviewer #1: Complement system is one of the first line innate immune defense mechanisms to confer the clearance of pathogens. Microbes have been documented to develop different strategies to escape from complement-mediated killing. In this study, Clark et al. described a novel complement evasion mechanism of Porphyromonas gingivalis (Pg) is one of the causative agents of periodontal diseases. The authors identified PG0352 as a sialidase to cleave different complement components, inactivating complement system, resulting in the survival of bacteria in the sera. Additionally, high resolution structure of PG0352 described in this manuscript correlates the structures and functions further pinpointing the two potential distinct functions conferred by two domains of this protein. This study was carefully performed, leading to the convincing results. The manuscript was well written and is easy to understand. I only have few suggestions for this manuscript:

Reviewer #2: This clearly presented study by Clark, Pham and coworkers entitled, “Functional and structural analyses reveal that a dual domain sialidase protects bacteria from complement killing through desialylation of complement factors,” describes the structural and functional characteristics of the protein, PG0352 of Porphyromonas gingivalis. The authors previously showed that a PG0352-defective Pg mutant was sensitive to complement. This mutant (for reasons that are unclear) produces less capsule and gingipains, which degrade complement factors. Hence, the complement resistance of the mutant could be due to any of several factors.

The authors show that (a) PG0352 is homologous to known sialidases and exhibits sialidase activity when tested using a model substrate; (b) PG0352 but not a polyalanine mutant PG0352AAAA that exhibits no activity, de-sialylates numerous complement/complement regulatory proteins, i.e., C1q, C4, C5, C9, FH, C4BP, and FI, in a time-dependent manner; (c) serum treated with the protein is 50% less able to kill E. coli K12. These data indicate that the impaired production of Pg capsule and gingipains do not need to be invoked to explain the observed complement resistance of Pg.

The authors also present structural studies that reveal two PG0352 domains: the C-terminal Sialidase domain exhibits sialic acid-binding and enzymatic activity at a wide pH range, while the non-catalytic N-terminal carbohydrate binding domain binds to glycosylated serum proteins. Structural analysis of six high resolution co-crystal structures of PG0352, revealed key residues involved in binding and in catalysis. Hence, PG0352 is a sialidase that may utilize the N-terminal carbohydrate binding domain domain to bind glycosylated serum proteins and positions associated sialic acid to be removed by the C-terminal sialidase domain.

Reviewer #3 (transcribed from jpg attachment provided): This is a well-written manuscript describing structural and functional analysis  of a sialidase of Porphyromonas gingivalis PG0352. The study is thorough, and the results convincing.  It is well known that one important virulence mechanism of Pg is its resistance to complement, but the underlying molecular mechanism was unknown.  This study fill this important gap.

**Part II – Major Issues: Key Experiments Required for Acceptance**

Reviewer #1: None

Reviewer #2: 1. Although several complement proteins are subject to the sialidase activity of recombinant PG0352, the biologically relevant target(s) of this activity is(are) not identified. De-sialylation of several complement proteins has been been investigated; in some cases, e.g. C1q and C1 inhibitor, there is no or little effect (https://pubmed.ncbi.nlm.nih.gov/6608959/, https://pubmed.ncbi.nlm.nih.gov/1468580/), in other cases (e.g. FH), activity has been reported to increase (https://pubmed.ncbi.nlm.nih.gov/6238095/) or decrease (https://pubmed.ncbi.nlm.nih.gov/33777036/). Indeed, although it seems perfectly plausible that the biological target is one or more of the complement proteins the authors identify, it remains possible that some other mechanism is at work.

2. Although the sialidase activity is required for de-sialylation of complement proteins by recombinant protein, the role of this activity in Pg is unexplored; a Pg mutant producing enzymatically inactive PG0352 is not characterized.

Reviewer #3: None

**Part III – Minor Issues: Editorial and Data Presentation Modifications**

Reviewer #1: 1. Complement proteins are present in the blood. Thus, complement evasion often occurs when pathogens come in contact with blood. Pg is often localized at the cavity of teeth. Occasionally, Pg may disseminate systematically. Therefore, would the PG0352-mediated complement evasion mostly occur during the systemic infection of Pg? The implications of the findings would need to be discussed.

2. Line 52-53, This manuscript is about the mechanisms of Pg in evading complement. Therefore, the C3 inhibitors as therapeutics to treat periodontitis may be not relevant to this work and may need to be removed.

3. Lin 99, OspE accounts for a group of proteins, and each Lyme borreliae species or strain differs in the number of copies of ospE family it encodes. That is, B. burgdorferi strains may have different number of CRASPs. Therefore, it is better to remove "three" in line 99.

4. The findings revealed by the newly solved high resolution structure of PG0352 is important. Such findings may be better to be summarized by a sentence and added at the end of the intro.

Reviewer #2: 3. The polyalanine mutant PG0352AAAA was included in the experiments of Figs. 1, 2, and 4A and B, but not Figs. 3 and 4C. It would be useful to include this control throughout.

4. The authors have shown that PG0352 removes sialic acid from complement proteins and immunoglobins using various assays systems in Figs. 1-4. Have any experiments been done to test whether PG0352 desialylates other proteins in plasma?

5. Have the authors determined if human serum contains antibody that recognizes PG0352?

6. Figure 2A-H. A. Full-length PG0352 desialylates multiple complement proteins including C4, C5, FH and C4BP. However, in Fig 8D, many of these proteins were not identified in the pull-down assay using CBM. Are there any experiments, possibly using another assay system, that tests full length (and CBM) PG0352 for binding to C4, C5, FH and C4BP? B. The bands for the marker cannot be seen; the authors can just label bands to the left of the blot. C. A degradation control for Fig. 2, like that performed for Fig. 1, should be included.

7. Fig. 3B-C, the font is difficult to read.

8. Fig. 4C shows a decrease in complement deposition when E. coli is treated with PG0352. The bands in the Coomassie gel of the cell lysate are faint for both conditions and the authors should consider densitometric analysis.

9. In the text, the authors provide a thorough description of key interactions between the sialidase domain and its substrates in Fig. 6A-D. However, identification of key interactions in these figures, as well as in Fig. 8F, should be included. In addition, a LIGPLOT in the supplemental information would be a great addition as this would provide a visualization of the detailed interactions of key residues.

10. Line 112. Clarify what is meant by "molecular mechanism". Mechanism of cleavage? Downstream effect of cleavage?

11. Line 136: “PG0352 construct in which residues Tyr-193….” Please clarify why were these residues chosen, e.g., a reference that highlights the importance of these residues. Does the mutant fold normally, e.g., by circular dichroism?

12. Line 184-185 and Fig.3. Have any studies addressed the activity of the truncated protein (PG0352CT) in di-sialylation of FH or C1q

13. In Line 232 and 233, the authors state RMSD values of 2.1 and 1.54 as exhibiting “strong structural homology”, could the authors clarify the statements? Was this compared to structures with RMSD values greater than 2.1?

14. Lines 246-255, describes in detail of the interaction between citrate and PG0352, but the data is not shown. Could a figure of this data be presented in the supplemental?

15. Fig. 8A, 8D and 8E. Label the molecular weights of reference bands on the lane of protein markers; Fig. 8D and 8E vs. Fig. 8 legend. “CMB” in the figures might be changed to “rCMB” to make consistent with legend.

16. Line 132-134, The authors used Sambucus nigra (SNA) lectin to specifically detect α-2,3- and α-2,6-linked sialic acids. Fig. 1 showed that several protein bands were detected in normal human serum treated with inactive PG0352 by SNA blot analysis. Are there other types of sialic acids in human serum, which can’t be recognized by SNA, but contribute to complement activation and can be de-sialylated by PG0352.

17. Line 466. It is clear to any reader that the structural studies were performed independently of the study of ref. 47. It is my opinion that the results of ref. 47 should be mentioned earlier in the manuscript, e.g., in the Results section.

Reviewer #3 (transcribed from jpg attachment provided):

1. It will be helpful to provide some background how the authors came about PG0352. Which strain did you use?  At first read, I was not sure if PG0352 was a strain name or something else. 

2.  The figure legends need more details. The full names of abbreviations in the figures and diagram should be spelled out in the legends, e.g., SNA, ConA, etc. The amounts  of proteins used should also be listed.  One should not need to read Methods to understand how the expt was done.

3. In Figure 1 & 2, SDS-PAGE or Western should be included to ensure the disappearing bands were not due to protease contaminations.

4. For mutant PG0352AAAA, what is the rationale to make Ala substitutions at these positions?  These positions, as well as other important residues, should be labeled in the structure in Figure 5.  The Neu5Ac coloring is not easy to discern--pink and red are difficult to differentiate.

5. Figure 6 is confusing. Consider only highlighting differences between ligands, as well as the PG0352 residues.

PLOS authors have the option to publish the peer review history of their article (what does this mean?). If published, this will include your full peer review and any attached files.

Reviewer #1: No

Reviewer #2: No

Reviewer #3: No
---

## [Decision Letter · Decision Letter 1]

7 Sep 2023

Dear Professor Li,

Thank you very much for submitting your manuscript "Functional and structural analyses reveal that a dual domain sialidase protects bacteria from complement killing through desialylation of complement factors" for consideration at PLOS Pathogens. As with all papers reviewed by the journal, your manuscript was reviewed by members of the editorial board and by several independent reviewers. The reviewers appreciated the attention to an important topic. Based on the reviews, we are likely to accept this manuscript for publication, providing that you modify the manuscript according to the review recommendations. To this point, Reviewer 2 has a single comment for you to address.  Once this item is addressed, please submit a revised version for final consideration.

Sincerely,

Jon T. Skare

Guest Editor

PLOS Pathogens

Karla Satchell

Section Editor

PLOS Pathogens

Kasturi Haldar

Editor-in-Chief

PLOS Pathogens

orcid.org/0000-0001-5065-158X

Michael Malim

Editor-in-Chief

PLOS Pathogens

orcid.org/0000-0002-7699-2064

Reviewer Comments (if any, and for reference):

Reviewer's Responses to Questions

**Part I - Summary**

Reviewer #1: (No Response)

Reviewer #2: strong study with both biochemical and structural components.

Reviewer #3: This is an excellent study. The revision has effectively addressed most critiques form the reviewers.

**Part II – Major Issues: Key Experiments Required for Acceptance**

Reviewer #1: (No Response)

Reviewer #2: None

Reviewer #3: None

**Part III – Minor Issues: Editorial and Data Presentation Modifications**

Reviewer #1: (No Response)

Reviewer #2: The authors should explain in the Results section leading up to Fig 4 the reasons why they cannot perform the experiment in PG rather than the E. coli surrogate. This might include issues of expression of the inactive mutant in PG or the inability to perform "extracellular" complementation by adding recombinant protein to PG prior to challenging with complement. Then, in the Discussion, the authors should comment on the unavoidable limitations of utilizing the E. coli surrogate. These additions will make it clear to readers that the authors have taken a systematic approach but one limited by technical challenges beyond their current control.

Reviewer #3: None

PLOS authors have the option to publish the peer review history of their article (what does this mean?). If published, this will include your full peer review and any attached files.

Reviewer #1: No

Reviewer #2: No

Reviewer #3: No

Figure Files:

Data Requirements:

Reproducibility:

References:

---

## [Editor Report · Decision Letter 2]

8 Sep 2023

Dear Professor Li,

We are pleased to inform you that your manuscript 'Functional and structural analyses reveal that a dual domain sialidase protects bacteria from complement killing through desialylation of complement factors' has been provisionally accepted for publication in PLOS Pathogens.

Best regards,

Jon T. Skare

Guest Editor

PLOS Pathogens

Karla Satchell

Section Editor

PLOS Pathogens

Kasturi Haldar

Editor-in-Chief

PLOS Pathogens

orcid.org/0000-0001-5065-158X

Michael Malim

Editor-in-Chief

PLOS Pathogens

orcid.org/0000-0002-7699-2064
---

## [Editor Report · Acceptance letter]

20 Sep 2023

Dear Professor Li,

We are delighted to inform you that your manuscript, "Functional and structural analyses reveal that a dual domain sialidase protects bacteria from complement killing through desialylation of complement factors," has been formally accepted for publication in PLOS Pathogens.

Best regards,

Kasturi Haldar

Editor-in-Chief

PLOS Pathogens

orcid.org/0000-0001-5065-158X

Michael Malim

Editor-in-Chief

PLOS Pathogens

orcid.org/0000-0002-7699-2064